# Modelling Attention with Aitchison Geometry: Token Distinguishability and Temperature Scaling

**Sam Hilton-Jones** [1]   **Timothy J. Norman** [1]   **Zhanxing Zhu** [1]

## Abstract

The attention mechanism with softmax normalisation is a foundational component of Transformer-based large language models. However, with very long contexts, attention scores are known to diminish, raising fundamental questions about token distinguishability and how it can be preserved. In this work, we provide a formal characterisation of token distinguishability in attention as a function of context length and embedding dimension. We introduce Aitchison distance to quantify relative differences among attention probabilities, and show that, with Gaussian queries and keys, even in the long-context regime, token distinguishability converges to a finite, non-zero limit rather than vanishing. Leveraging the linear relationship between inverse-temperature scaling and Aitchison distance, we derive a theoretical lower bound of $\Omega(\sqrt{\log L})$ on the logit scaling required to produce a sharp attention distribution. Finally, we demonstrate that Aitchison distance provides a principled and practical alternative to entropy for monitoring training and inference, as it captures the full compositional structure, including the smaller components of the attention probabilities.

## 1. Introduction

As a key building block in Transformer-based large language models (LLMs), the attention mechanism is used to model the relations between tokens in a sequence (Bahdanau et al., 2015; Kim et al., 2017; Vaswani et al., 2017), where a softmax operation normalises the attention logits to a probability distribution, i.e. summing to one. However, for models with *very long contexts*, the softmax inevitably shrinks the indi-

vidual probabilities as the number of tokens increases (Chi et al., 2024; Yin et al., 2024; Nakanishi, 2025; Vasylenko et al., 2026). This phenomenon is termed as "*vanishing attention*" (Mudarisov et al., 2025) or "*softmax dispersion*" (Veličković et al., 2025), which has raised a significant concern regarding LLMs' selective ability among tokens, i.e. *token distinguishability*, when dealing with long contexts.

A commonly used strategy to overcome vanishing attention is to use temperature scaling on the logits before the softmax. The inverse-temperature function increases with the context length $L$, resulting in a sharper output distribution. The order and nature of this scaling function have come under much scrutiny (Yao et al., 2021; Jianlin, 2021; Chiang & Cholak, 2022; Peng et al., 2024; Chi et al., 2024; Zhang et al., 2024; Veličković et al., 2025; Vasylenko et al., 2026), with a logarithmic scaling factor $\log L$ being practically implemented in numerous models (Bai et al., 2023; Nakanishi, 2025; Puvvada et al., 2025).

Two concurrent works, Giorlandino & Goldt (2026) and Chen et al. (2026), theoretically investigated the inverse-temperature scaling function from a phase transition perspective, deriving $\sqrt{\log L}$ and $\log L$ scaling, respectively, based on different simplified models and assumptions. That is, below a critical value of the suggested scaling, the attention scores tend towards uniformity due to the dispersion of the attention mass between an increasing number of tokens, while for larger scaling, they are highly saturated, leading to entropy collapse (See more discussions in Section 4).

However, a formal and comprehensive modelling of token distinguishability is still underexplored, and how it influences temperature scaling is not clarified. Concretely, two key questions remain outstanding: *1) In relative terms, how does the token distinguishability within the attention distribution change with context length and embedding dimension? 2) What is the exact relationship between the temperature scaling and token distinguishability?*

To answer these questions, we propose a modelling of the distinguishability of two tokens, $i$ and $j$, with the *log ratio of their attention probabilities*, i.e. $\log \frac{p_i}{p_j}$, theoretically studying how this ratio changes when varying the context length and embedding dimension. Based on this, we theo-

---

[1]School of Electronics and Computer Science, University of Southampton, Southampton, United Kingdom. Correspondence to: Sam Hilton-Jones <s.hiltonjones@soton.ac.uk>, Zhanxing Zhu <z.zhu@soton.ac.uk>.

*Proceedings of the 43rd International Conference on Machine Learning*, Seoul, South Korea. PMLR 306, 2026. Copyright 2026 by the author(s).

retically derive a lower bound for the inverse-temperature scaling function with respect to context length, thus guiding its practical usage.

The novelty lies in that we introduce a tool from Compositional Data Analysis (CoDA), **Aitchison distance** (Aitchison, 2003), to compare the attention distribution and the uniform distribution (indicating no distinguishability between tokens). Aitchison distance has two key properties: **1)** By measuring the log ratios, this factors out the magnitude reduction of the probability components with increasing context length, unlike Euclidean distance; and **2)** It considers all probability components, including the small probabilities, rather than a focus on the larger components as seen with entropy measures, elaborated in Section 2.

We derive the probability density function (PDF) of the Aitchison distance to uniform, finding that the expectation of the Aitchison distance is non-vanishing with context length, concluding that *the log ratios of the probabilities do not vanish with context length*. Therefore, in relative terms and despite the magnitudes decreasing, the attention probabilities are equally as distinguishable with longer contexts, up to machine precision, elaborated in Corollary 3.3.

While this distinguishability is preserved with infinite precision, in the real world, underflowing to $0$ is a real concern when using very long contexts. Therefore, considering the linear relationship between Aitchison distance and inverse-temperature scaling, we derive a lower bound on the inverse-temperature scaling function for the expected $\log$ maximum probability to be constant with context length, see Proposition 4.1. In particular, we show that maintaining the expected $\log$ maximum probability above a constant is not possible without an inverse-temperature scaling function of order $\Omega(\sqrt{\log L})$. Hence, while our theoretical investigation studies the idealistic conditions of infinite machine precision and infinite context length, we provide theoretically-grounded practical guidance for the real-world conditions where we would want to avoid token uniformity by suggesting temperature scaling.

Finally, we present the practical use of Aitchison distance as a measure to monitor the training and inference process, informing about the token distinguishability of real-world models. In particular, when using Aitchison distance to compare a base and length-extrapolated model, OpenLLaMA-3B (Geng & Liu, 2023; Together Computer, 2023; Touvron et al., 2023a) and LongLLaMA-3B (Tworkowski et al., 2023), we find more focused attention within "memory layers", reorganising the hierarchical structure of the lesser weighted tokens, a property that KL is less sensitive towards.

Our contributions are as follows:

- We introduce Aitchison distance as a new tool to measure the distance to uniformity of the attention probabilities, finding that the token distinguishability is non-vanishing with the increasing context length.

- We statistically investigate the relation between inverse-temperature scaling and token distinguishability, providing a statistical lower bound of $\Omega(\sqrt{\log L})$ order scaling to hold the expected $\log$ maximum probability constant.

- We demonstrate the use of Aitchison distance as a measure during training and inference, informing about the token distinguishability between context lengths in large-scale models.

### Related Work

**Signal Propagation** Signal propagation within Transformers has been a widely studied area. In particular, failure modes such as "representational collapse" (Barbero et al., 2024) and "rank collapse", the inductive bias of self-attention to *token uniformity* (Dong et al., 2021), are studied extensively, with many works considering tokens as particles to study this collapse to uniformity (Geshkovski et al., 2023; 2024; Karagodin et al., 2024; Geshkovski et al., 2025; Cowsik et al., 2025; Bruno et al., 2025; Chen et al., 2025; Polyanskiy et al., 2025; Rigollet, 2025; Alcalde et al., 2025). This collapse has been shown to cause a hindrance to training by causing the query and key gradients to vanish (Noci et al., 2022). Similarly, Zhai et al. (2023) described "entropy collapse" as the reduction in the entropy of the attention head, correlating with training instability. Building on this, Hong & Lee (2025) demonstrated that softmax attention probabilities' entropy is strictly decreasing with the logits' variance. Therefore, while we derive a lower bound for the inverse-temperature scaling function, entropy collapse due to high logit variance suggests that this function can not be increased arbitrarily.

**Shrinking Attention** Closely related to our study, many works have noted the reduction of the attention probabilities magnitude with an increasing context length (Yan et al., 2020). Veličković et al. (2025) showed the inability of attention models to generalise a sharp function in-distribution to out-of-distribution longer context examples, coining *"softmax dispersion"*. Likewise, Mudarisov et al. (2025) has formalised the "softmax bottleneck" (Yang et al., 2018), such that the Euclidean distance between a top-$N$ selected relevant tokens and noise tokens shrinks with context length. The softmax function is largely blamed for this behaviour, suggesting that the probability distribution becomes increasingly flat as the number of tokens grows (Nakanishi, 2025). Similarly, Richter & Wattenhofer (2020) has studied the token distinguishability, stating that a model has an intrinsic focus on the local information from the residual connection.

**Overcoming Vanishing Attention** Within long context models, the "overallocation" of attention mass on irrelevant context has been identified as a bottleneck (Ye et al., 2025). Therefore, the suppression of the smaller probability components is a key feature of "focused" Transformers such as LongLLaMA (Tworkowski et al., 2023), a property that we explore further in Section 5.

Temperature scaling can be used to increase the sharpness of the attention probabilities, which is commonly investigated within the literature (Yao et al., 2021; Chiang & Cholak, 2022; Bai et al., 2023; Peng et al., 2024; Chi et al., 2024; Zhang et al., 2024; Veličković et al., 2025; Nakanishi, 2025; Puvvada et al., 2025; Vasylenko et al., 2026). The theoretical basis for the order of the inverse temperature has also been investigated; Jianlin (2021) and Chen et al. (2026) find a $\log L$ order scaling to hold the entropy and the angle between tokens constant, respectively, whereas Giorlandino & Goldt (2026) finds a $\sqrt{\log L}$ scaling to match self-attention to the Random Energy Model (REM). Therefore, the order of the inverse-temperature scaling function found depends on the modelling assumptions and the objective of the scaling of the logits. In this work, we use Gaussian queries and keys to precisely model the token distinguishability within the probabilities, linking this to the inverse-temperature scaling and deriving a lower bound for a constant expected $\log$ maximum probability.

While these works have discussed the bias towards token uniformity during training and the shrinking of attention probabilities with context length, a rigorous investigation into the relationship between the log ratios of attention probabilities and context length remains unclear. Likewise, the precise relation of temperature scaling with token distinguishability remains underexplored. To analyse these relations, we utilise Aitchison distance, a measure that has not been considered when studying attention uniformity within the attention probabilities, despite its strength in analysing sequences that have a finite sum.

## 2. Aitchison Geometry for Attention

We aim to precisely characterise the relative structure of the attention probabilities. Attention probabilities always sum-to-one, living on the simplex $\mathscr{S}^{L-1} = \{(x_1, \ldots, x_L) : x_1 \geq 0, \ldots x_L \geq 0; x_1 + \cdots + x_L = 1\}$. Therefore, the magnitude of the probability for each token depends on the context length, with longer sequences having smaller individual components. To analyse the distinguishability between probabilities, a relative comparison of the attention probabilities that ignores the inevitable shrinkage in the magnitudes of the probability components is required.

To this end, we consider the log ratios of attention probabilities, $\log \frac{p_i}{p_j}$, which are invariant to global rescaling and

renormalisation. We use the centred log-ratio (clr) transformation $\mathrm{clr}(\mathbf{x}) = [\ln \frac{x_1}{g(\mathbf{x})}, \ldots, \ln \frac{x_L}{g(\mathbf{x})}]$, where $g(\mathbf{x}) = \left(\prod_{i=1}^{L} x_i\right)^{1/L}$ is the geometric mean, giving a vector in the zero-sum space $H = \{\mathbf{z} \in \mathbb{R}^L : \mathbf{1}^\top \mathbf{z} = 0\}$. This transformation encodes the pairwise log ratios as the differences of the clr coordinates, i.e. $\mathrm{clr}(\mathbf{p})_i - \mathrm{clr}(\mathbf{p})_j = (\log p_i - \overline{\log \mathbf{p}}) - (\log p_j - \overline{\log \mathbf{p}}) = \log \frac{p_i}{p_j}$, where $\overline{\log \mathbf{p}} = \frac{1}{L} \sum_{i=1}^{L} \log p_i$.

Therefore, on the zero-sum space, the inverse of the clr is the softmax function $\texttt{softmax}(\cdot) : H \to \mathscr{S}^{L-1}$. Hence, the clr of the attention probabilities is equivalent to the centred attention logits $\bar{\mathbf{a}} = \mathbf{a} - \frac{1}{L} \sum_{i=1}^{L} a_i \mathbf{1}$, where $\mathbf{a}$ is the attention logits.

**Aitchison geometry** (Aitchison, 1982; 2003) formalises this log ratio viewpoint by building a vector space structure on the simplex and transforming Euclidean geometry to the clr space. Within this geometry, the distance between $\mathbf{x}, \mathbf{y} \in \mathscr{S}^{L-1}$ is known as the Aitchison distance given as

$$d_A(\mathbf{x}, \mathbf{y}) = \sqrt{\frac{1}{2L} \sum_{i=1}^{L} \sum_{j=1}^{L} \left(\ln \frac{x_i}{x_j} - \ln \frac{y_i}{y_j}\right)^2}.$$

This distance is a metric on the simplex and depends only on the log ratios and relative magnitudes of the components of the vectors. Additionally, the neutral element of the vector space is the uniform vector $\mathbf{u} = [\frac{1}{L}, \cdots, \frac{1}{L}]$, giving $\mathrm{clr}(\mathbf{u}) = 0$.

To measure the flatness of the attention mechanism, we analyse the Aitchison distance between the attention probabilities $\mathbf{p}$ and the uniform distribution $\mathbf{u}$, denoted as $d_A(\mathbf{p}, \mathbf{u})$, which can be computed simply with

$$d_A(\mathbf{p}, \mathbf{u}) = \|\mathrm{clr}(\mathbf{p}) - \mathrm{clr}(\mathbf{u})\| = \|\mathrm{clr}(\mathbf{p})\| = \|\bar{\mathbf{a}}\|,$$

where this property is particularly useful when implementing Aitchison distance in real-world models by avoiding the calculation of the $\log$ of a small probability.

We use this distance comparison due to the uniform distribution being the degenerate case, i.e. no token distinguishability, therefore providing a baseline for the worst case form of token distinguishability. The Aitchison distance to uniform measures the root-mean-square of the log ratio differences, equivalent to the norm of the centred attention logits $\bar{\mathbf{a}}$, a simple and easily implementable measure. Based on this, the Aitchison distance will grow as the length of $\mathbf{p}$ grows, due to this being the norm of more log elements. Therefore, to allow the use of Aitchison distance to compare different context lengths, we use **normalised Aitchison distance** as $\bar{d}_A(\mathbf{x}, \mathbf{y}) = \frac{1}{\sqrt{L}} \cdot d_A(\mathbf{x}, \mathbf{y})$. This removes the $\sqrt{L}$ growth due to this being the $L_2$ norm of more elements, and allows the comparison between context lengths.

Additionally, Aitchison distance to uniform establishes the *linear application of inverse-temperature scaling*, i.e. with inverse-temperature function $T$, $\|\bar{\mathbf{a}}\| \mapsto T \cdot \|\bar{\mathbf{a}}\| = d_A(\mathbf{p}_T, \mathbf{u})$, where $\mathbf{p}_T$ represents the attention probability with inverse-temperature $T$ applied. Hence, applying an inverse-temperature linearly impacts the Aitchison distance, motivating our investigation into the relationship between vanishing attention and variable temperature scaling using Aitchison geometry, see more details in Section 4.

**Comparison to Entropy Measures** Despite not being metrics, entropy-based measures such as the KL divergence to uniformity are often popular choices to consider the peakiness of a distribution. For example, entropy is used to consider the peakiness of a distribution within attention heads in Zhai et al. (2023), linking this to training stability.

However, when considering the flatness of the attention probabilities, entropy-based measures such as KL divergence, $\mathrm{KL}(\mathbf{p}\|\mathbf{u})$, mainly focus on larger elements. Take Figure 1, demonstrating the contour level sets on a three-part simplex, with the following points: $\mathbf{p}_1$ - a dominant component, sub-dominant element and small component, and $\mathbf{p}_2$ - one dominant component and two smaller but similar components, despite the structure of the smaller components changing, KL divergence to uniform is approximately unchanged, whereas Aitchison distance does consider these small component changes. Therefore, while entropy measures are less sensitive to changes in smaller components, Aitchison distance measures the change equally along the full vector, retaining sensitivity in smaller components.

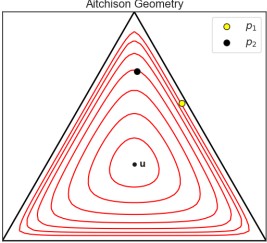 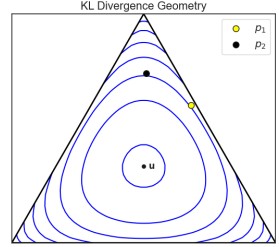

*Figure 1.* Level sets around the uniform point on a three-part simplex for Aitchison distance and KL divergence, demonstrating the retention of sensitivity for the smaller components that Aitchison distance maintains, unlike KL. The points $\mathbf{p}_1 = [0.02, 0.38, 0.6]$ and $\mathbf{p}_2 = [0.12, 0.14, 0.74]$ demonstrate this, such that $\mathrm{KL}(\mathbf{p}_1\|\mathbf{u}) \approx \mathrm{KL}(\mathbf{p}_2\|\mathbf{u})$ but $d_A(\mathbf{p}_1, \mathbf{u}) > d_A(\mathbf{p}_2, \mathbf{u})$.

Within the attention mechanism, this tail behaviour remains important, particularly for long contexts. Since attention probabilities are propagated through many layers, small changes in the tail components can have a large impact on the model's final output, particularly when these act on large or aligned value vectors. Therefore, by considering Aitchison distance, we can measure the change in the full probability vector within the attention mechanism.

One such example occurs when comparing OpenLLaMA-

3B (Geng & Liu, 2023; Together Computer, 2023; Touvron et al., 2023a) with its length-extrapolated version, LongLLaMA-3B (Tworkowski et al., 2023). Within the "memory layers" of this model, the lesser weighted tokens in the tail are hierarchically reorganised; moderately weighted tokens are up-weighted, and very small weighted tokens are down-weighted. This increases the Aitchison distance to uniformity, which is sensitive to small probability components, whereas KL divergence is relatively insensitive to these changes. We study this closer within Section 5, visualising the $\log$ of the attention probabilities of LongLLaMA-3B and OpenLLaMA-3B in Figure 5.

## 3. Analysis of Aitchison Distance

Based on the relations between the Aitchison distance and token distinguishability described in Section 2, we present an analytical investigation into the statistical properties of the Aitchison distance to the uniform distribution of the attention probabilities, showcasing its first and second moments relationship with context length and embedding dimension.

**Attention Mechanism** To analyse the statistical properties of the attention probabilities, we consider the query and key vectors with the token representations $\mathbf{x}$ as $\mathbf{q}_i = \mathbf{x}_i W_Q$ and $\mathbf{k}_i = \mathbf{x}_i W_K$, with $W_Q, W_K \in \mathbb{R}^{d \times d}$, where $d$ is the head embedding dimension, and we use $T(L)$ as an inverse-temperature function dependent on the context length $L$. Therefore, we have the attention probabilities as

$$p_{ij} = \frac{\exp(T(i) \cdot \frac{\mathbf{q}_i \mathbf{k}_j^\top}{\sqrt{d}} + m_{ij})}{\sum_{\ell=1}^{L} \exp(T(i) \cdot \frac{\mathbf{q}_i \mathbf{k}_\ell^\top}{\sqrt{d}} + m_{i\ell})},$$

where $i$ is the available context length of the current query, and $m_{ij}$ is an element of the standard causal mask $M \in \mathbb{R}^{L \times L}$, i.e. $m_{ij} = 0$ for $j \leq i$ and $m_{ij} = -\infty$ otherwise, making this an autoregressive mechanism.

Our primary assumption is that the query and key vectors can be assumed to be independent and normally distributed. In this way, the query and key vectors are i.i.d. and distributed as $\mathbf{q} \sim \mathcal{N}(\mathbf{0}, \phi_q^2 I_d)$, $\mathbf{k}_i \sim \mathcal{N}(\mathbf{0}, \phi_k^2 I_d)$, where $\phi_q^2, \phi_k^2$ are the variance of the normal distribution, assumed to be equal, i.e. $\phi_q^2 = \phi_k^2 = \phi_{qk}^2$. This model setup and assumptions have been widely accepted within the literature due to its tractability (Lee et al., 2018; Nahshan et al., 2024).

**Distribution of Aitchison Distance** We are interested in deriving the statistical properties of the Aitchison distance between the attention probabilities $\mathbf{p}$ and uniform vector $\mathbf{u}$ to measure token distinguishability; hence, we denote $\tilde{a}$ as

$$\tilde{a} = d_A(\mathbf{p}, \mathbf{u}) = \|\operatorname{clr}(\mathbf{p})\| = \|\log \mathbf{p} - \frac{1}{L}\sum_{i=1}^{L} \log p_i \mathbf{1}\|.$$

We use a mixture distribution model of a Gaussian distribution with Gamma distributed variance to state the following in Proposition 3.1.

**Proposition 3.1.** *(**Probability Density Function of** $\tilde{a}$) With Gaussian distributed query and key vectors* $\mathbf{q}, \mathbf{k}_i \in \mathbb{R}^{1 \times d}$, *the PDF of the attention probabilities Aitchison distance to uniform* $\tilde{a}$ *can be expressed as*

$$f_{\tilde{a}}(x) := \frac{1}{2^{\frac{d+L-5}{2}} c^{\frac{d+L-1}{4}} \Gamma(\frac{L-1}{2}) \Gamma(\frac{d}{2})} x^{\frac{d+L-3}{2}}$$
$$\cdot K_{\frac{d-L+1}{2}} \left( \frac{x}{\sqrt{c}} \right), \tag{1}$$

*where* $K_\nu(\cdot)$ *is the modified Bessel function of the second kind, and* $c = \frac{\phi_{qk}^4}{d}$.

**Lognormal Approximation** Due to the complex form of this PDF and prior works' investigation into the distribution of the attention matrix, we compare this distribution with the distribution produced if the probabilities were approximated with a lognormal distribution, following the characterisation within Nahshan et al. (2024). Hence, for the vector $\mathbf{y} \sim \text{Lognormal}(\mu\mathbf{1}, \sigma_1^2 I_L)$, the PDF of the norm of the clr vector, $\hat{a} = \|\text{clr}(\mathbf{y})\|$, is given as

$$f_{\hat{a}}(x) = \frac{2^{1-\frac{L-1}{2}}}{\Gamma(\frac{L-1}{2})\sigma_1^{L-1}} x^{L-2} \exp\left( -\frac{x^2}{2\sigma_1^2} \right), \tag{2}$$

where the derivation of this form is given in Appendix A.4.

The distribution of the norm of the clr attention probabilities with Gaussian queries and keys and the transformed lognormal probabilities are visualised in Figure 2, along with the theoretical PDFs in Proposition 3.1 and Equation 2, demonstrating the closeness between the lognormal proxy and the attention probabilities with Gaussian queries and keys.

To compare these two distributions of the norm of the clr, we find the optimal shape parameter $\sigma_1$ to maximise the cosine similarity between the two distributions. We find that $\sigma_1^2 \to 1$ as the head's embedding dimension becomes infinitely large. More details for this derivation can be found in Appendix A.4. While we do not study the use case of the lognormal distribution as a linear attention kernel (Nahshan et al., 2024), we anticipate that future investigations continue to consider this distribution as a suitable proxy.

**Attention Distance to Uniformity** The main use case for the PDF derived in Proposition 3.1 is to determine its moments as a function of context length $L$ and embedding dimension $d$. Further, the moments quantify the distribution of the Aitchison distance of the attention probabilities to

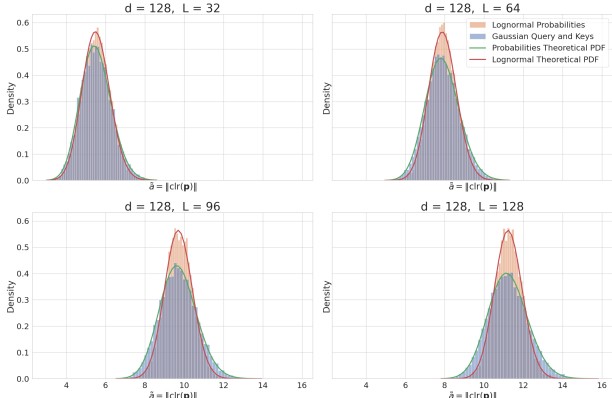

*Figure 2.* Histogram of $\|\text{clr}(\mathbf{p})\|$, with $\mathbf{p}$ generated by the softmax of Gaussian query and keys, and by Lognormal($\mu\mathbf{1}$, $\sigma_1^2 I_L$), with $\sigma_1^2 = 1$, and the theoretical PDFs in Proposition 3.1 and Equation 2, for a head dimension $d = 128$ and variable context length $L$.

the uniform distribution. The moment generating function (MGF) of $\tilde{a} = \|\text{clr}(\mathbf{p})\|$ is presented in Corollary 3.2.

**Corollary 3.2.** *(**Moment Generating Function of** $\tilde{a}$) The MGF of* $\tilde{a} = \|\text{clr}(\mathbf{p})\|$ *with Gaussian query and key vectors* $\mathbf{q}, \mathbf{k}_i \in \mathbb{R}^{1 \times d}$ *is given by*

$$M_{\tilde{a}}(t) = \frac{2^d c^{-\frac{d}{2}} (\frac{1}{\sqrt{c}} - t)^{-d} \Gamma(\frac{d+1}{2}) \Gamma(\frac{L}{2})}{\sqrt{\pi}}$$
$$\cdot {}_2\tilde{F}_1 \left( d, \frac{d-L+2}{2}; \frac{d+L}{2}; 1 + \frac{2}{\sqrt{c} \cdot t - 1} \right), \tag{3}$$

*for* $t < \frac{1}{\sqrt{c}}$, *where* ${}_2\tilde{F}_1(\cdot)$ *is the regularised hypergeometric function.*

Due to the rapid growth of this function, visualised in Appendix A.2, the higher moments of this distribution are much larger, suggesting that there is a significant proportion of mass that can be positioned away from the mean. This means that the attention probabilities at initialisation can regularly deviate from uniformity.

Using this MGF, we can derive the moments of this distribution. These moments describe the Aitchison distance to uniform; therefore, with our definition of the normalised Aitchison distance $\bar{d}_A(\mathbf{p}, \mathbf{u})$, we derive moments about the following components

$$\bar{d}_A(\mathbf{p}, \mathbf{u})^2 = \text{Var}_i(\log p_i)$$
$$= \text{Var}_i(a_i)$$
$$= \frac{1}{2} \mathbb{E}_{i,j} \left[ \left( \log \frac{p_i}{p_j} \right)^2 \right], \tag{4}$$

where $p_i$ and $p_j$ are uniformly sampled pairs, and we verify these relations in Appendix A.3.

The relationship between the normalised Aitchison distance and the log ratios demonstrates the intuition behind this measure for token distinguishability. If the expected log ratios between tokens are large, and therefore the variance of the log probabilities is large, then tokens have large differences between each other. Hence, when the normalised Aitchison distance is large, the token distinguishability is large, justifying this measure.

Considering these relations, and the aim to characterise the attention structure as a function of context length and embedding dimension, we take the first two moments from the MGF and consider the limit in the infinite context length regime in Corollary 3.3.

**Corollary 3.3.** *(Non-vanishing token distinguishability)*
*The infinite context length limit of the expectation of the normalised Aitchison distance is given as*

$$\lim_{L \to \infty} \mathbb{E}[\bar{d}_A(\mathbf{p}, \mathbf{u})] = \frac{\sqrt{2c} \cdot \Gamma(\frac{d+1}{2})}{\Gamma(\frac{d}{2})}, \qquad (5)$$

*and of the normalised squared Aitchison distance as*

$$\lim_{L \to \infty} \mathbb{E}[\bar{d}_A(\mathbf{p}, \mathbf{u})^2] = \frac{2c \cdot \Gamma(\frac{d+2}{2})}{\Gamma(\frac{d}{2})} = cd. \qquad (6)$$

*Therefore, with Gaussian query and key vectors, the expectation of the normalised Aitchison distance has a non-zero, finite limit.*

Hence, the expectation of the standard deviation of the log probabilities does not shrink to $0$ in the limit. If this were not the case, the $\sqrt{\text{Var}_i(\log p_i)}$ would converge to $0$ in the limit, stating no variability in the $\log$ attention probability vectors as the context length grows.

In previous works, it has been found that the Euclidean distance between the relevant subset of token representations and noise tokens shrinks as the context length increases (Mudarisov et al., 2025). However, when considering a measure that looks past the inevitable $\frac{1}{L}$ magnitude shrinkage in individual components, such as Aitchison distance, the relative structure of the attention probabilities does not shrink with context length, instead converging to a finite, non-zero constant and retaining distinguishability up to machine precision. Therefore, our findings do not contradict those of Mudarisov et al. (2025), but instead model the relative structure of the components of the attention probabilities rather than the absolute magnitude distance of the context vector for relevant tokens.

## 4. Statistical View of Temperature Scaling

A simple and effective way to avoid the vanishing attention phenomenon is the use of an inverse-temperature function scaling with context length to increase the magnitude of the attention logits. While there does exist research investigating the critical scale of this temperature function (Jianlin, 2021; Giorlandino & Goldt, 2026; Chen et al., 2026), we consider the scaling from the point of view of the probability distribution derived in Section 3. In particular, with the softmax of Gaussian query and keys, we derive a necessary lower bound for the inverse-temperature scaling function to enable a constant expected $\log$ maximum probability in Proposition 4.1.

**Proposition 4.1.** *(Statistical Temperature Scaling)* *With Gaussian query and key vectors, forming the attention probabilities* $\mathbf{p}$, *with a maximum component* $p_{\max} = \max_i p_i$, *if* $\mathbb{E}[\log p_{\max}] \geq \log C, \quad \forall L > 1$, *where $C$ is a constant, then necessarily the inverse-temperature scaling function is* $T(L) = \Omega(\sqrt{\log L})$.

*Proof Sketch:* To prove this lower bound, we apply the following logic: taking $\mathbb{E}[\log p_{\max}] \geq \log C$ as a lower bound for the expectation of the maximum $\log$ probability, where $C$ is a constant. If we derive an upper bound as a function dependent on context length and temperature scaling, such that $\mathbb{E}[\log p_{\max}] \leq f(L, T(L))$, then it is clear that $f(L, T(L)) \geq \log C$, as the upper bound must be greater than the lower bound. We use the bound on the magnitude of $\mathbb{E}[\tilde{a}]$, with a large context length, to derive this upper bound. Then, we can solve this inequality for $T(L)$, the inverse-temperature scaling function, giving a necessary condition. We highlight that this is a necessary condition for the lower bound condition $\mathbb{E}[\log p_{\max}] \geq \log C$ to be possible, not a sufficient condition.

The full statistical derivation of this can be found in Appendix A.5.

**Comparison with Concurrent Works** The scaling function required for a non-vanishing attention probability is closely linked to the concurrent works deriving a $\sqrt{\log L}$ (Giorlandino & Goldt, 2026) and $\log L$ (Chen et al., 2026) order scaling. The main differences between these works are the imposition of the attention logits being correlated Gaussian variables, mapping self-attention to the REM, seen in Giorlandino & Goldt (2026). Chen et al. (2026) discusses these differences, conducting an additional short heuristic derivation for the $\sqrt{\log L}$ scale, given independent Gaussian attention logits. In contrast to this, we take independent Gaussian query and key vectors to derive a precise modelling of the distribution of the attention logits at initialisation. Therefore, we have a fine-grained view of the distribution of the attention logits, enabling our direct link to Aitchison geometry, and motivating a new measure to describe token distinguishability during training and inference. Additionally, this distribution is used to derive a lower bound for the expected $\log$ maximum probability to be constant with context length, and therefore the attention probability being non-vanishing.

Another key difference is that, in these works, a model over the token representations after the full attention mechanism is proposed. Contrastingly, we specifically model and investigate the attention probabilities before any value embedding. We make this choice to formally investigate the *softmax dispersion* discussed within Veličković et al. (2025), and probability bounds within Mudarisov et al. (2025), where the softmax is deemed responsible for the failures of long context tasks through the reduction in attention mass.

## 5. Aitchison Distance as a Measure

In the previous sections, we have considered the key properties that the Aitchison distance of the attention probabilities can describe. Based on this, we use the normalised Aitchison distance during the training of real-world models on an NLP task to investigate the following: **1)** How does temperature scaling impact the token distinguishability over the course of training? **2)** How do different training regimes influence the token distinguishability within the model?

Additionally, to motivate the advantage of Aitchison distance over KL, we investigate at inference time the shift in the probability components after length extrapolating a model. As a major difference in finetuning for length-extrapolated models is the suppression of smaller probability components, Aitchison distance is a particularly informative measure, while KL is less sensitive to these changes. For this comparison, we use **normalised KL divergence** $\overline{\text{KL}(\mathbf{p}\|\mathbf{u})}$ such that we divide by the maximum KL divergence possible, $\log L$, to produce a divergence between 0 and 1, and allow the comparison between different context lengths.

For each of these experiments, the implementation details and parameters used are described within Appendix B.2. Given the definition of the Aitchison distance, it is clear that this measure becomes problematic when attempting to take the $\log$ of 0. However, as discussed, the normalised Aitchison distance is equivalent to the norm of the centred attention logits divided by $\sqrt{L}$. Therefore, when practically implementing this distance, we can measure the distance before the softmax is applied and instead calculate through the attention logits for the current context length, avoiding the difficulties of taking the $\log$ of small probabilities and enhancing the practical implementation benefits of Aitchison distance.

**Token Distinguishability with Temperature Scaling** As seen above, the normalised Aitchison distance describes the relative ratios of the tokens within the attention probabilities. Therefore, we use this property to study how the relative ratios within a real-world model, Pythia-1B (Biderman et al., 2023), that we pretrain on the NLP dataset, RedPajama-Data-1T-Sample 1B (Together Computer, 2023), change

during training with different temperature scales. In particular, we use the inverse-temperature scaling functions

$$T(L) = \beta \cdot \{1, \sqrt{\log L}, \log L, (\log L)^2\},$$

where $\beta$ is a learnable parameter for each layer, and has been passed through a ReLU function to ensure positivity, i.e. $\beta = \text{ReLU}(\beta_{\text{raw}})$, initialised to $\beta_0 = 1.0$ for all layers, alike to the exploratory initialisation within Nakanishi (2025). We log the normalised Aitchison distance and normalised KL divergence to uniform for Row 8 and Row 2048 of the causal attention matrix, averaged over each head and micro-batch forward-passed through the model within each layer. We plot these distances, along with the standard deviation across the 16 layers, with the training loss in Figure 3.

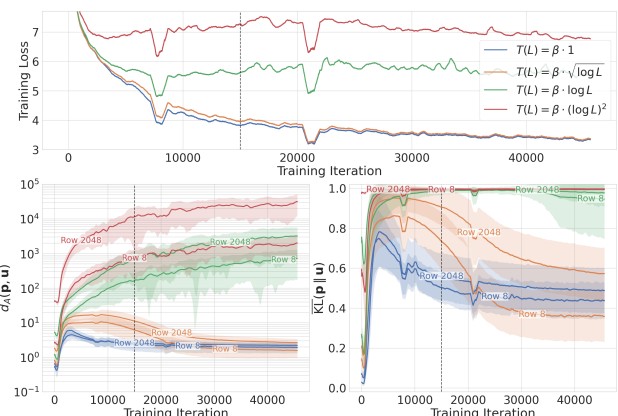

*Figure 3.* Training loss, normalised Aitchison distance, and normalised KL divergence to uniform for Row 8 and Row 2048 of the attention probabilities of a Pythia-1B model trained on the RedPajama-Data-1T-Sample 1B dataset, each with an inverse-temperature scaling function $T(L) = \beta \cdot \{1, \sqrt{\log L}, \log L, (\log L)^2\}$, where $\beta$ is initialised to $\beta_0 = 1.0$, with the vertical dashed line separating the saturation period for $T(L) = \log L$ and $T(L) = (\log L)^2$ at 15,000 iterations.

We first note in Figure 3 the difference between the measurements of the Aitchison distance and KL divergence. These two measures consider different properties; Aitchison distance measures the average log ratios, whereas KL focuses on the sharpness of the distribution. As observed in Figure 3, for $T(L) = \log L$ and $T(L) = (\log L)^2$, KL suggests that the sharpness is at full saturation after 15,000 iterations, whereas the continued growth of Aitchison distance indicates that the smaller probability components are getting even smaller, producing more token distinguishability.

By investigating the change in the token distinguishability with temperature scaling we make the following observations: **1)** Token distinguishability scaling between rows with inverse-temperature function is preserved during training, such that the order in which the logits are scaled match with the final token distinguishability order, i.e. $T(L) = (\log L)^2$ has the largest difference between

Row 2048 and Row 8's normalised Aitchison distance, whereas $T(L) = 1$ has the smallest difference between these rows, indicating a higher selectivity at longer contexts compared to short contexts with a higher ordered inverse-temperature function. **2)** A too large scaling can unstabilise training, as seen with the normalised Aitchison distance with $T(L) = (\log L)^2$ and $T(L) = \log L$ scaling, indicating a commonly saturated softmax. This reflects the entropy collapse noted in prior works, as the temperature scale has proportionally increased the variance of the attention logits, thereby increasing training loss (Zhai et al., 2023; Hong & Lee, 2025). **3)** As anticipated with NLP tasks, a more selective attention pattern is required at longer context lengths, hence the token distinguishability is larger in Row 2048 compared to Row 8 for all models.

Finally, we note that in Figure 3, we are interested in the difference in token distinguishability between the short-context (Row 8) and long-context (Row 2048) causal-attention rows. The magnitude of these rows is largely dependent on the initialised scale of $\beta_0$, which is not the focus of our investigation. Therefore, to verify that our findings on the difference between rows hold with different initialisation scales, we conduct the same experiment as Figure 3 with a $\beta_0 = 0.5$ initialisation, and with $\beta_0$ initialised to the reciprocal of the average inverse-temperature scaling function's value during training (Nakanishi, 2025), i.e. $\beta_0 = \frac{2048}{\sum_{n=1}^{2048} T(n)}$, in Appendix B.3.

**Token Distinguishability with Training Regimes** Moving beyond the temperature scaling impact on token distinguishability, we now consider how token distinguishability changes as a result of the training regime used. Again, we can investigate this due to the properties of the Aitchison distance, tracking the log ratios of the probabilities, hence informing about token distinguishability. We train three Pythia-1B models, each with a $T(L) = \beta \cdot \sqrt{\log L}$ inverse-temperature scaling function: a Healthy model (standard hyperparameters), Underfitting 1 (small learning rate and high weight decay), and Underfitting 2 (high learning rate and no weight decay). The full parameter settings used are given in Appendix B.2. We train each of these on the RedPajama-Data-1T-Sample 1B dataset to plot the normalised Aitchison distance, normalised KL divergence to uniform with the standard deviation across layers and training loss in Figure 4

From Figure 4, we observe that, while the average normalised Aitchison distance at the end of training is similar between the Healthy model and Underfitting 1, the variance of this distance between various layers of the model is larger in the Healthy model, suggesting that many layers of the Healthy model are more selective than others, producing a more diverse range of token distinguishability. Conversely, Underfitting 1 has a small variance, suggesting

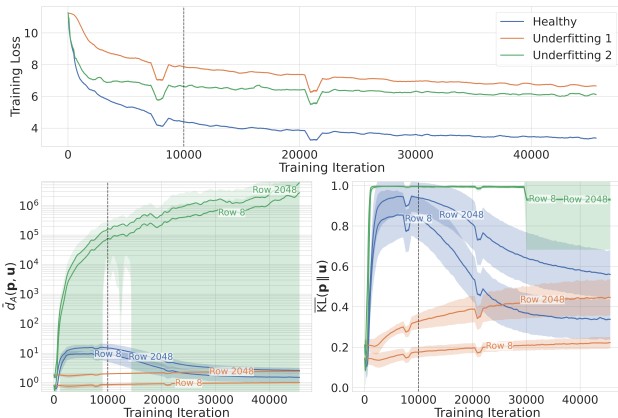

*Figure 4.* Training loss, normalised Aitchison distance, and normalised KL divergence to uniform for Row 8 and Row 2048 of the attention probabilities of three Pythia-1B models trained on the RedPajama-Data-1T-Sample 1B dataset with $T(L) = \beta\sqrt{\log L}$. These models are: A Healthy model with standard hyperparameters, Underfitting 1 with strong weight decay and a small learning rate, and Underfitting 2 with no weight decay and a high learning rate. The vertical dashed line represents the saturation point of many of the attention probabilities of Underfitting 2 at 10,000 iterations.

that this model has a more constrained selectivity across layers, producing a higher training loss. Hence, we can draw the following insights: **1)** Underfitting an NLP model by having a small learning rate and high regularisation can constrain the layer-wise variation of the normalised Aitchison distance, producing a more common attention pattern, however, a too aggressive training regime with high learning rate and small regularisation promotes the saturation of the attention probabilities resulting in a rapidly increasing normalised Aitchison distance within many layers, with a large variation in the extent of this saturation across layers. In NLP tasks, both local and longer-range attention patterns can be promoted; therefore, the model that has a range of selectivity patterns, the Healthy model, performs better than the underfitting model that has little variation of selectivity patterns, and the model with layers either being saturated or complete uniformity, with $\beta = 0.0$, seen in Underfitting 2 at 30,000 iterations. **2)** The difference in selectivity between the rows, i.e. the difference in $\bar{d}_A(\mathbf{p}, \mathbf{u})$ between short context rows (Row 8) and long context rows (Row 2048), is strongly influenced by the training regime. A too large difference between rows, as seen with Underfitting 2, likely indicates unstable training. **3)** An overly aggressive training regime can result in an exploding Aitchison distance, hence producing mostly saturated attention probabilities and failing to learn.

**Token Distinguishability at Inference** As highlighted within Section 2, the key difference between Aitchison distance and KL divergence is the sensitivity to measuring the changes in the smaller probability components. A key ex-

ample of where we must consider the smaller probability components shift is within length-extrapolated models. As the context length increases, many irrelevant tokens receive a small but non-zero attention mass, thereby distracting from more relevant tokens (Ye et al., 2025; Vasylenko et al., 2026). Hence, length-extrapolated models redistribute this attention from the smaller components towards the relevant tokens, thereby suppressing the smaller probabilities.

One such example of this is within the length-extrapolated version of OpenLLaMA-3B (Geng & Liu, 2023; Together Computer, 2023; Touvron et al., 2023a), LongLLaMA-3B (Tworkowski et al., 2023). LongLLaMA uses "memory layers" in Layers 6, 12 and 18, such that each query in the memory layers "attends to preceding keys from the local context and the top $k$ most matching keys (i.e. having the largest inner product with the query) from memory", producing more "focused" attention, with higher token distinguishability.

We aim to investigate the shift of attention probability distribution between the length-extrapolated model and the base model. In Figure 5, we plot the following: in the first row, the pooled ridge plot of the $\log p_i$ with various context lengths for both models; in the second row, an entropy weighted ridge plot of $\log p_i$, where each sample is weighted to by $-p_i \log p_i$, highlighting the $\log p_i$ values that contribute the most towards the entropy; and in the third row, we plot the change in the normalised Aitchison distance and change in the normalised KL divergence per context length between the two models, i.e. $\bar{d}_A(\mathbf{p}_{\text{LongLLaMA}}, \mathbf{u}) - \bar{d}_A(\mathbf{p}_{\text{OpenLLaMA}}, \mathbf{u})$, and $\overline{\text{KL}}(\mathbf{p}_{\text{LongLLaMA}} \| \mathbf{u}) - \overline{\text{KL}}(\mathbf{p}_{\text{OpenLLaMA}} \| \mathbf{u})$.

Taking Layer 6 as an example, the change in the $\log p_i$ components indicates that there are many more small components; likewise, there is also an increase in the moderately sized probability components. Hence, this layer is more focused, having a larger proportion of the attention mass on a smaller subset of tokens, suppressing the smaller probability tokens further, and increasing the variance in the $\log$ probabilities. Despite this, the normalised Aitchison distance and normalised KL divergence disagree on the direction of uniformity between the models. Aitchison distance suggests that this is a less uniform and higher token distinguishability distribution within LongLLaMA, while KL suggests that this is a more uniform distribution, highlighting the key difference between the measures. The ridge plot of the entropy contributions, visualised in the second row, indicates that *entropy is mostly unaware of the change and suppression of the smaller probability components, mainly considering the shift in the larger components*, due to the rescaling of entropy by $-p_i \log p_i$. Therefore, to accurately track the changes within the full probability vector, including the smaller components, Aitchison distance is a suitable mea-

sure choice, with this being a useful tool to consider the distribution shift in length-extrapolated models. We take a further example of Llama-2-7B (Touvron et al., 2023b) and its length-extrapolated model Llama-2-7B-32k (Together AI, 2023) in Appendix B.4.2.

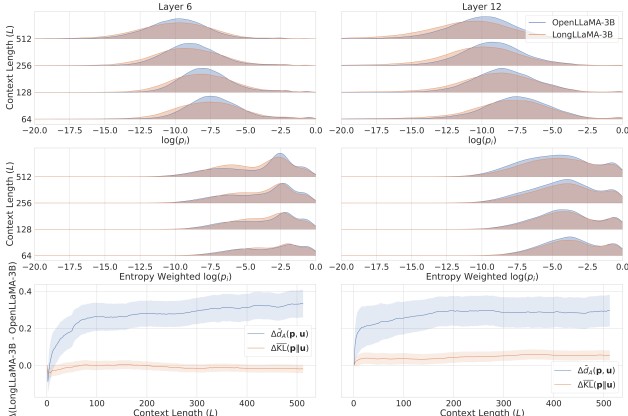

*Figure 5.* First Row: Ridge plot, showcasing the histogram of $\log p_i$ for an increasing context length of OpenLLaMA-3B and LongLLaMA-3B (Layer 6, 12). Second Row: Entropy weighted $\log(p_i)$ ridge plot, weighting each sample with $-p_i \log p_i$, showing its contribution towards entropy measures. Third Row: The difference $\Delta$ between LongLLaMA-3B and OpenLLaMA-3B's normalised Aitchison distance and normalised KL divergence to uniform over the context length, averaged over all heads and over 10 random prompts of length 512 from the RedPajama-Data-1T-Sample 1B dataset with the standard error.

## 6. Discussion and Conclusion

In this work, we have shown that while vanishing attention describes the reduction in the attention probabilities magnitude due to the finite-sum constraint, the relative difference between components does not collapse due to context length. Thus, a long context sequence does not automatically imply uniform attention. Further, our analysis, using Aitchison distance, of the lower bound of the inverse-temperature scaling function for a constant expected $\log$ maximum probability provides a concrete link between token distinguishability and logit scaling.

By introducing Aitchison distance as a measure during training and inference, we reveal its strength in describing the full attention probabilities, with the use of Aitchison distance to consider the hierarchical shift of more "focused" models, demonstrating its advantages. Additionally, the introduction of this distance measure is strongly linked with failure modes such as "entropy collapse" (Zhai et al., 2023; Hong & Lee, 2025), when tracking real-world models.

In the future, we can envisage the characterisation of Aitchison distance in different contexts, such as after stacking many attention layers. Likewise, we can extend our theoretical analysis of token distinguishability past initialisation, investigating the Aitchison distance during training.

## Acknowledgements

This work was supported by the UK Research and Innovation Centre for Doctoral Training in Machine Intelligence for Nano-Electronic Devices and Systems [EP/S024298/1].

The authors acknowledge the insightful discussions with Prof. Patrick Rebeschini regarding the key proofs. The authors also acknowledge the use of the IRIDIS HPC and associated support services at the University of Southampton in the completion of this work.

## Impact Statement

This paper presents work whose goal is to advance the field of Machine Learning. There are many potential societal consequences of our work, none of which we feel must be specifically highlighted here.

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

The Appendix contains two parts. Appendix A contains the derivations and equations used within this work. Appendix B contains the implementation details, empirical expansions and verifications, along with further comparisons between length-extrapolated models' attention probability distribution shift.

## A. Theoretical Derivations

Within this section, we derive the key equations, relations and bounds for the conclusions made within the paper. We start with the derivation of the probability density function (PDF) in Section A.1, followed by the derivation of the moment generating function (MGF) and first and second moments with their limit in Section A.2. Next, we verify the relations between the log ratios, the Aitchison distance and statistical properties of the attention probabilities in Section A.3. The full derivation of the closed-form for the cosine similarity between the lognormal proxy for the attention probabilities, along with the optimisation problem for the shape parameter $\sigma_1$, is found in Section A.4. Finally, the lower bound for the inverse-temperature scaling function for a constant expected $\log$ maximum probability $\mathbb{E}[\log p_{\max}]$ is within Section A.5.

### A.1. Attention Scores Distribution Derivation

First, we derive the PDF for the norm of the centred attention logits, or Aitchison distance, given in Proposition 3.1.

Note that we derive this distribution without temperature scaling. To start, we have the following vector for the attention logits of available context length $L$

$$\mathbf{a}_L = \frac{1}{\sqrt{d}}[\mathbf{qk}_1^\top, \mathbf{qk}_2^\top, \ldots, \mathbf{qk}_L^\top], \quad \mathbf{q}, \mathbf{k}_i \in \mathbb{R}^{1 \times d}.$$

Due to the assumption that the query and key vectors are independent Gaussian vectors, the query and each key vector have a distribution given by

$$\mathbf{q} \sim \mathcal{N}(\mathbf{0}, \phi_{qk}^2 I_d), \quad \mathbf{k}_i \sim \mathcal{N}(\mathbf{0}, \phi_{qk}^2 I_d).$$

We condition on $\mathbf{q}$, which is used with all of the keys, $\mathbf{k}_1, \ldots, \mathbf{k}_L$, to state the conditional distribution as

$$\mathbf{a}_L | \mathbf{q} \sim \mathcal{N}\left(\mathbf{0}, \frac{\|\mathbf{q}\|^2 \phi_{qk}^2}{d} I_L\right).$$

Hence, we can write this as a mixture representation as

$$\mathbf{a}_L = \sqrt{V}\mathbf{Z}_L, \quad \mathbf{Z}_L \sim \mathcal{N}(\mathbf{0}, I_L), \quad V \sim \frac{\phi_{qk}^4}{d}\chi_d^2,$$

where $\mathbf{Z}_L$ and $V$ are independent, we have a chi-squared distribution as a result of the squared norm of a Gaussian vector, and $\frac{1}{d}$ due to the normalisation factor $\frac{1}{\sqrt{d}}$ applied to the attention logits.

Therefore, we have a mixture distribution given as a Gaussian distribution with a Gamma-distributed variance. Hence, we have two PDFs as

$$f_V(v) = \frac{1}{2^{\frac{d}{2}}\Gamma(\frac{d}{2})c^{\frac{d}{2}}}v^{\frac{d}{2}-1}\exp(\frac{-v}{2c}), \quad \text{where} \quad c = \frac{\phi_{qk}^4}{d},$$

$$f_{\mathbf{a}_L|V}(\mathbf{a}\,|\,v) = \frac{1}{(2\pi v)^{\frac{L}{2}}}\exp(-\frac{\|\mathbf{a}\|^2}{2v}),$$

marginalising over $v$ gives the following PDF

$$f_{\mathbf{a}_L}(\mathbf{a}) = \frac{1}{2^{\frac{d}{2}-1}c^{\frac{d+L}{4}}\Gamma(\frac{d}{2})(2\pi)^{\frac{L}{2}}}\|\mathbf{a}\|^{\frac{d-L}{2}}K_{\frac{d-L}{2}}(\frac{\|\mathbf{a}\|}{\sqrt{c}}),$$

where $K_\nu(\cdot)$ is the modified Bessel function of the second kind.

This is the distribution of the attention logits. To transform this to be the Aitchison distance, we derive the distribution of the $\mathrm{clr}(\mathbf{p})$. This vector is within a lower-dimensional space given as

$$H = \{\mathbf{x} \in \mathbb{R}^L, \mathbf{1}^\top \mathbf{x} = 0\},$$

such that $\mathrm{clr}(\mathbf{p}) \in H$ has $L-1$ dimensions. Hence, projecting our distribution into $H$, by centring the distribution, results in the distribution with $\mathbf{x} \in H$ given as

$$f_{\mathrm{clr}(\mathbf{p})}(\mathbf{x}) = \frac{1}{2^{\frac{d}{2}-1} c^{\frac{d+L-1}{4}} \Gamma(\frac{d}{2})(2\pi)^{\frac{L-1}{2}}} \|\mathbf{x}\|^{\frac{d-L+1}{2}} K_{\frac{d-L+1}{2}}\left(\frac{\|\mathbf{x}\|}{\sqrt{c}}\right). \tag{7}$$

We can note that the distribution of $\mathrm{clr}(\mathbf{p})$ is the projection of a spherically symmetric distribution, hence the resultant distribution is also spherically symmetric, an important property that we utilise in deriving the lower bound of the inverse-temperature scaling.

Finally, we take the standard transformation from the vector distribution to the radial distribution (Gupta & Song, 1997) to have the PDF expressed as the following

$$f_{\tilde{a}}(x) = \frac{1}{2^{\frac{d+L-5}{2}} c^{\frac{d+L-1}{4}} \Gamma(\frac{L-1}{2})\Gamma(\frac{d}{2})} x^{\frac{d+L-3}{2}} K_{\frac{d-L+1}{2}}\left(\frac{x}{\sqrt{c}}\right), \tag{8}$$

with $x \geq 0$, matching the form given in Proposition 3.1.

## A.2. MGF, Moments, and Limits of $\tilde{a}$

Given the PDF derived above, the calculation of the MGF is given by the integral

$$M_{\tilde{a}}(t) = \mathbb{E}[\exp(t\tilde{a})] = \int_0^\infty \frac{1}{2^{\frac{d+L-5}{2}} c^{\frac{d+L-1}{4}} \Gamma(\frac{L-1}{2})\Gamma(\frac{d}{2})} x^{\frac{d+L-3}{2}} K_{\frac{d-L+1}{2}}\left(\frac{x}{\sqrt{c}}\right) \cdot \exp(tx) \, dx.$$

Using integral 6.621.3 from Gradshteyn & Ryzhik (2007), we can derive the MGF as

$$M_{\tilde{a}}(t) = \frac{2^d c^{-\frac{d}{2}} (\frac{1}{\sqrt{c}} - t)^{-d} \Gamma(\frac{d+1}{2})\Gamma(\frac{L}{2})}{\sqrt{\pi}} \cdot {}_2\tilde{F}_1\left(d, \frac{d-L+2}{2}; \frac{d+L}{2}; 1 + \frac{2}{\sqrt{c} \cdot t - 1}\right), \tag{9}$$

where ${}_2\tilde{F}_1(\cdot)$ is the regularised hypergeometric function and $t < \frac{1}{\sqrt{c}}$. We plot the function $M_{\tilde{a}}(\frac{t}{\sqrt{L}})$ to remove the dimensional growth effects and demonstrate the rapid growth of the function in Figure 6.

We can use this function to obtain all of the moments of $\tilde{a}$, whose first two moments are

$$\mathbb{E}[\tilde{a}] = \frac{2\sqrt{c} \cdot \Gamma(\frac{d+1}{2}) \cdot \Gamma(\frac{L}{2})}{\Gamma(\frac{d}{2}) \cdot \Gamma(\frac{L-1}{2})}, \tag{10}$$

$$\mathbb{E}[\tilde{a}^2] = \frac{4c \cdot \Gamma(\frac{d+2}{2}) \cdot \Gamma(\frac{L+1}{2})}{\Gamma(\frac{d}{2}) \cdot \Gamma(\frac{L-1}{2})}. \tag{11}$$

As we can use the relation $\mathbb{E}[\tilde{a}] = \mathbb{E}[d_A(\mathbf{p}, \mathbf{u})]$ to consider the change in the Aitchison distance with context length, we now want to consider the asymptotic limit of this expectation with respect to $L$. Therefore, using the asymptotic identity for the ratio of $\Gamma(\cdot)$ functions, the expectation is given by

$$\mathbb{E}[\tilde{a}] \sim \frac{2\sqrt{c} \cdot \Gamma(\frac{d+1}{2})}{\Gamma(\frac{d}{2})} \cdot \sqrt{\frac{L}{2}},$$

which we can transform to normalised Aitchison distance by dividing by the dimensional $\sqrt{L}$ growth, giving the limit as

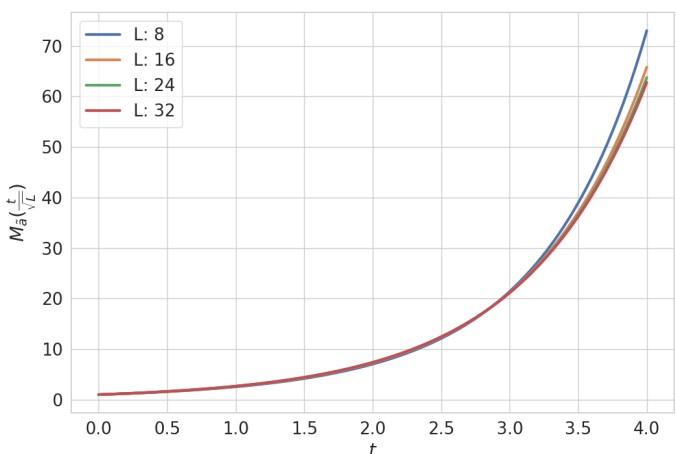

*Figure 6.* Plot of $M_{\bar{a}}(\frac{t}{\sqrt{L}})$, with $d = 32$, demonstrating the growth with $t$ and the convergence as the context length $L$ increases towards a common function.

$$\lim_{L\to\infty} \mathbb{E}[\bar{d}_A(\mathbf{p}, \mathbf{u})] = \frac{\sqrt{2c} \cdot \Gamma(\frac{d+1}{2})}{\Gamma(\frac{d}{2})}. \tag{12}$$

Likewise, we derive a similar limit for the square of the normalised Aitchison distance as

$$\lim_{L\to\infty} \mathbb{E}[\bar{d}_A(\mathbf{p}, \mathbf{u})^2] = \frac{2c \cdot \Gamma(\frac{d+2}{2})}{\Gamma(\frac{d}{2})} = cd. \tag{13}$$

Therefore, the expected normalised Aitchison distance at initialisation has a non-zero finite limit, and the attention probabilities do not collapse to a flat uniform vector, in expected relative terms.

### A.3. Verification of the relations in Equation 4

In this section, we verify the following relations

$$\bar{d}_A(\mathbf{p}, \mathbf{u})^2 = \mathrm{Var}_i(\log p_i) = \mathrm{Var}_i(a_i) = \frac{1}{2}\mathbb{E}_{i,j}\left[\left(\log\frac{p_i}{p_j}\right)^2\right], \tag{14}$$

where the variance is across the coordinates, and the expectation is over independent uniformly random indices $i, j \in [1, L]$.

First, the definition of the normalised Aitchison distance is given as

$$\bar{d}_A(\mathbf{p}, \mathbf{u}) = \frac{1}{\sqrt{L}}\|\operatorname{clr}(\mathbf{p}) - \operatorname{clr}(\mathbf{u})\|_2 = \frac{1}{\sqrt{L}}\|\operatorname{clr}(\mathbf{p})\|_2,$$

due to $\mathbf{u}$ being the neutral element and having a $\operatorname{clr}$ transform of $0$.

Substituting the $\operatorname{clr}$ transform of $\mathbf{p}$ gives

$$\frac{1}{\sqrt{L}}\|\operatorname{clr}(\mathbf{p})\|_2 = \frac{1}{\sqrt{L}}\sqrt{\sum_{i=1}^{L}(\log p_i - \bar{\ell})^2}, \quad \text{where} \quad \bar{\ell} = \frac{1}{L}\sum_{i=1}^{L}\log p_i.$$

In this notation, we define $\mathrm{Var}_i(\log p_i)$ as the coordinate-wise variance as

$$\mathrm{Var}_i(\log p_i) = \frac{\sum_{i=1}^{L}(\log p_i - \bar{\ell})^2}{L}.$$

Therefore, we have the relation that

$$\bar{d}_A(\mathbf{p}, \mathbf{u})^2 = \text{Var}_i(\log p_i).$$

Further, the following identity for the variance of the log probabilities is given by

$$\text{Var}_i(\log p_i) = \text{Var}_i\left(\log \frac{\exp a_i}{\sum_k \exp(a_k)}\right) = \text{Var}_i\left(\log \exp(a_i) - \log\left(\sum_k \exp(a_k)\right)\right).$$

Now, as the $\log\left(\sum_k \exp(a_k)\right)$ term is constant for all indices of this probability vector, and therefore does not change the variance, this is simply given as

$$\text{Var}_i(\log p_i) = \text{Var}_i(a_i).$$

Finally, we consider the relationship of $\mathbb{E}_{i,j}\left[\left(\log \frac{p_i}{p_j}\right)^2\right]$ to these functions. We have the following expression

$$\mathbb{E}_{i,j}\left[\left(\log \frac{p_i}{p_j}\right)^2\right] = \mathbb{E}_{i,j}[(\log p_i - \log p_j)^2] \tag{15}$$

$$= \mathbb{E}_{i,j}[((\log p_i - \bar{\ell}) - (\log p_j - \bar{\ell}))^2] \tag{16}$$

$$= \frac{1}{L^2} \sum_{i=1}^{L} \sum_{j=1}^{L} (\log p_i - \log p_j)^2,$$

As the cross term disappears when summing over $i, j$ as $\sum_{i=1}^{L}(\log p_i - \bar{\ell}) = 0$, the double sum reduces to

$$\mathbb{E}_{i,j}\left[\left(\log \frac{p_i}{p_j}\right)^2\right] = \frac{1}{L^2} \sum_{i=1}^{L} \sum_{j=1}^{L}\left(\log \frac{p_i}{p_j}\right)^2 = \frac{2}{L} \sum_{i=1}^{L}(\log p_i - \bar{\ell})^2 = 2\,\text{Var}_i(\log p_i).$$

This verifies all of the relations.

### A.4. Lognormal Similarity

In this section, we compare the distribution of the norm of the clr probabilities, where the probabilities are approximated as a lognormal distribution, and as the softmax of the Gaussian query and key distribution stated in Proposition 3.1.

To compare the lognormal approximation of the attention probabilities, we first construct the PDF of the norm of the centred attention logits under this approximation.

We define the vector $\mathbf{y} \in \mathbb{R}^L$, lognormally distributed as

$$\mathbf{y} \sim \text{Lognormal}(\mu\mathbf{1}, \sigma_1^2\, I_L).$$

To map this distribution to the norm of the clr distribution, we use the projection

$$P = I_L - \frac{1}{L}\mathbf{1}\mathbf{1}^\top,$$

where $\mathbf{1}\mathbf{1}^\top$ is the $L$ dimensional matrix of ones. Hence, we perform the transformation given as $\hat{a} = \|P\log(\mathbf{y})\|$, matching this to the clr transformation.

Based on this, we can define $\mathbf{z} = \log \mathbf{y}$, given by the distribution

$$\mathbf{z} \sim \mathcal{N}(\mu\mathbf{1}, \sigma_1^2\, I_L).$$

Therefore, the distribution of $\hat{a}$ is the Euclidean norm of a centred projection of a normal distribution, which can be given by the following PDF

$$f_{\hat{a}}(x) = \frac{2^{1-\frac{L-1}{2}}}{\Gamma(\frac{L-1}{2})\sigma_1^{L-1}} x^{L-2} \exp\left(-\frac{x^2}{2\sigma_1^2}\right). \tag{17}$$

We verify this PDF by overlaying this theoretical distribution over the empirical estimates that we use in Section 3, Figure 2.

**Cosine Similarity**  To compare this distribution to the derived distribution for the Gaussian query and key vectors, we measure the cosine similarity between the distributions, given as

$$S_C(A, B) = \frac{A \cdot B}{\|A\|\|B\|} = \frac{\int_0^\infty f_A(x) f_B(x) dx}{\sqrt{\int_0^\infty f_A(x)^2 dx}\sqrt{\int_0^\infty f_B(x)^2 dx}},$$

where $f_A(x) = f_{\tilde{a}}(x)$ is the K-distribution family PDF for the norm of the $\mathrm{clr}(\mathbf{p})$ found in the above derivation, and $f_B(x) = f_{\hat{a}}(x)$ is the PDF of the norm of the $\mathrm{clr}$ transformed lognormal vector.

To evaluate these integrals, we use the special integral functions given in Gradshteyn & Ryzhik (2007). Hence, our expression for the cosine similarity is given by the following

$$S_C = \frac{D_1 D_2 \overbrace{\int_0^\infty x^{\frac{d+3L-7}{2}} \exp(-\frac{x^2}{2\sigma_1^2}) K_{\frac{d-L+1}{2}}(\frac{x}{\sqrt{c}}) dx}^{F_2}}{D_1 \cdot \sqrt{\underbrace{\int_0^\infty x^{d+L-3} K_{\frac{d-L+1}{2}}(\frac{x}{\sqrt{c}}) K_{\frac{d-L+1}{2}}(\frac{x}{\sqrt{c}}) dx}_{F_1}} \cdot D_2 \cdot \sqrt{\underbrace{\int_0^\infty x^{2L-4} \exp(-\frac{x^2}{\sigma_1^2}) dx}_{F_3}}},$$

with the constant prefactors that cancel as

$$D_1 = \frac{1}{2^{\frac{d+L-5}{2}} c^{\frac{d+L-1}{4}} \Gamma(\frac{L-1}{2}) \Gamma(\frac{d}{2})},$$

$$D_2 = \frac{2^{1-\frac{L-1}{2}}}{\Gamma(\frac{L-1}{2}) \sigma_1^{L-1}}.$$

Starting with $F_1$, we have the following integral

$$F_1 = \int_0^\infty x^{d+L-3} K_{\frac{d-L+1}{2}}(\frac{x}{\sqrt{c}}) K_{\frac{d-L+1}{2}}(\frac{x}{\sqrt{c}}) dx,$$

which, using integral 6.576.4, where $\lambda = -d - L + 3$, $\mu = \nu = \frac{d-L+1}{2}$, and $a = b = \frac{1}{\sqrt{c}}$, gives the following expression

$$F_1 = \frac{2^{d+L-5} c^{\frac{d+L-2}{2}}}{\Gamma(d+L-2)} \cdot \Gamma(\frac{d+L-2}{2})^2 \cdot \Gamma(\frac{2d-1}{2}) \cdot \Gamma(\frac{2L-3}{2}).$$

Next, for $F_2$, the integral is defined as

$$F_2 = \int_0^\infty x^{\frac{d+3L-7}{2}} \exp(-\frac{x^2}{2\sigma_1^2}) K_{\frac{d-L+1}{2}}(\frac{x}{\sqrt{c}}) dx, \tag{18}$$

which, using the integral 6.631.3, with $\mu = \frac{d+3L-7}{2}$, $\alpha = \frac{1}{2\sigma_1^2}$, $\nu = \frac{d-L+1}{2}$, and $\beta = \frac{1}{\sqrt{c}}$, reduces to

$$F_2 = 2^{\frac{d+3L-11}{4}} \exp(\frac{\sigma_1^2}{4c}) \sigma_1^{\frac{d+3L-7}{2}} \sqrt{c} \cdot \Gamma(\frac{d+L-2}{2}) \cdot \Gamma(\frac{2L-3}{2}) W_{-\frac{d+3L-7}{4}, \frac{d-L+1}{4}}(\frac{\sigma_1^2}{2c}),$$

where $W_{\mu,\nu}(x)$ is the Whittaker function.

Finally, $F_3$ is given by the integral

$$F_3 = \int_0^\infty x^{2L-4} \exp(-\frac{x^2}{\sigma_1^2}) dx.$$

Using integral 3.326.2, with $m_1 = 2L - 4$, $\beta = \frac{1}{\sigma_1^2}$, and $n = 2$, we have the expression

$$F_3 = \frac{\Gamma(\frac{2L-3}{2}) \sigma_1^{2L-3}}{2}.$$

Therefore, we can put this together to find the cosine similarity as

$$S_C = 2^{\frac{L-d+1}{4}} \cdot c^{\frac{4-d-L}{4}} \cdot \sigma_1^{\frac{d+L-4}{2}} \cdot \exp(\frac{\sigma_1^2}{4c}) \cdot \sqrt{\frac{\Gamma(d+L-2)}{\Gamma(\frac{2d-1}{2})}} \cdot W_{-\frac{d+3L-7}{4}, \frac{d-L+1}{4}}(\frac{\sigma_1^2}{2c}). \tag{19}$$

To empirically check this cosine similarity, we construct empirical PDFs from random samples and compare their cosine similarity with the theoretical expression in Equation 19. The resultant comparison, as a function of context length, is shown in Figure 7.

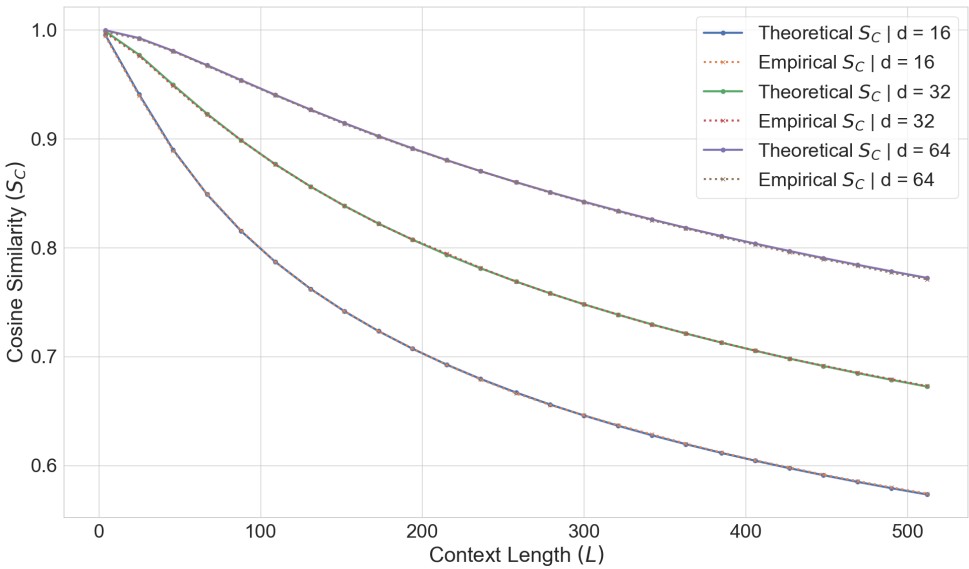

*Figure 7.* Empirical and theoretical cosine similarity, with $d = \{16, 32, 64\}$, and $L \in [2, 512]$, between the PDFs within Equation 8 and Equation 17.

We now find the stationary points of this cosine similarity. Therefore, we take the derivative of this function with respect to the shape parameter $\sigma_1$ to have the following expression

$$\frac{d}{d\sigma_1} \left( \sigma_1^{\frac{d+L-4}{2}} \cdot \exp(\frac{\sigma_1^2}{4c}) \cdot W_{-\frac{d+3L-7}{4}, \frac{d-L+1}{4}}(\frac{\sigma_1^2}{2c}) \right) = 0.$$

Hence, taking this derivative gives

$$\frac{1}{2c} \cdot \exp(\frac{\sigma_1^2}{4c}) \cdot \sigma_1^{\frac{1}{2}(-6+d+L)} \cdot \left( (c(2d+4L-11) + 2\sigma_1^2) W_{-\frac{d+3L-7}{4}, \frac{d-L+1}{4}}(\frac{\sigma_1^2}{2c}) \right.$$
$$\left. - 4cW_{-\frac{d+3L-11}{4}, \frac{d-L+1}{4}}(\frac{\sigma_1^2}{2c}) \right) = 0.$$

The prefactor for this expression is positive for $c > 0$, and $\sigma_1 > 0$, giving the equation to solve as

$$\left( c(2d+4L-11) + 2\sigma_1^2 \right) W_{-\frac{d+3L-7}{4}, \frac{d-L+1}{4}}(\frac{\sigma_1^2}{2c}) - 4cW_{-\frac{d+3L-11}{4}, \frac{d-L+1}{4}}(\frac{\sigma_1^2}{2c}) = 0.$$

To solve this, we take the recurrence relations of the Whittaker function in 13.15.11 (DLMF) with the following parameters

$$\kappa = -\frac{d+3L-7}{4}, \quad \mu = \frac{d-L+1}{4}, \quad z = \frac{\sigma_1^2}{2c},$$

and shape the expression to fit the known recurrence definition as

$$\frac{-2L+3}{4} W_{\kappa,\mu}(z) + (\kappa - \mu - \frac{1}{2})(\kappa + \mu - \frac{1}{2}) W_{\kappa-1,\mu}(z) = 0.$$

Rearranging this, we have the equation to solve as

$$\frac{W_{\kappa-1,\mu}(z)}{W_{\kappa,\mu}(z)} = \frac{1}{d+L-2}.$$

To evaluate the left-hand side, we simplify this expression as a continued fraction, and we use a Tricomi function ratio as

$$\frac{W_{\kappa-1,\mu}(z)}{W_{\kappa,\mu}(z)} = \frac{U(\frac{1}{2}+\mu-\kappa+1, 1+2\mu, z)}{U(\frac{1}{2}+\mu-\kappa, 1+2\mu, z)}.$$

Using the relation 13.3.7 from DLMF (DLMF), the function $y_n = U(a+n, b, z)$ satisfies a three-term recurrence relation of the form

$$y_{n+1} + B_n y_n + A_n y_{n-1} = 0,$$

where we have the definitions

$$y_n = U(a+n, b, z), \quad A_n = \frac{1}{(a+n)((a+n)-b+1)}, \quad B_n = \frac{(b-2(a+n)-z)}{(a+n)((a+n)-b+1)},$$

with

$$a = \frac{1}{2} + \mu - \kappa, \quad b = 1 + 2\mu.$$

For this recurrence, the sequence $y_n = U(a+n, b, z)$ is the minimal solution (Deaño & Temme, 2009), therefore, we can apply Pincherle's theorem (Elaydi, 2005) to give the ratio in the continued fraction form

$$\frac{y_{n+1}}{y_n} = -\frac{A_{n+1}}{B_{n+1} + \cfrac{-A_{n+2}}{B_{n+2} + \cfrac{-A_{n+3}}{B_{n+3}+\cdots}}}.$$

Hence, the left-hand side is given by the expression

$$\frac{U(a+1, b, z)}{U(a, b, z)} = -\frac{A_1}{B_1 + \frac{-A_2}{B_2+\cdots}} = \frac{\frac{-4}{(d+L)(2L-1)}}{-\frac{2(2z+d+3L-3)}{(d+L)(2L-1)} + \frac{\frac{-4}{(d+L+2)(2L+1)}}{-\frac{2(2z+d+3L+1)}{(d+L+2)(2L+1)}+\cdots}}.$$

Therefore, our new equation to solve is given by the following

$$\frac{\beta(1)}{\alpha(1)+} \frac{\beta(2)}{\alpha(2)+} \cdots = \frac{1}{d+L-2},$$

where this is the continued fraction with the parameters

$$\alpha(n) = -\frac{2(c(4n+d+3L-7)+\sigma_1^2)}{c(2n+d+L-2)(2n+2L-3)},$$

$$\beta(n) = -\frac{4}{(2n+d+L-2)(2n+2L-3)}.$$

While this is not an expression that is easily solvable, the continued fraction can be truncated to an arbitrary precision to give a polynomial that is easily solvable. We plot the solutions of the continued fraction with 20 continued fraction terms in Figure 8, to demonstrate how the shape parameter changes with context length.

By considering how the shape parameter changes in the infinite embedding-dimension limit, with context length fixed, we can analyse the leading order of each coefficient of the continued fraction. With $c = \frac{1}{d}$, we have the leading orders: $\beta(n) = \mathcal{O}(\frac{1}{d})$ and $\alpha(n) = \mathcal{O}(1)$, resulting in the tail of the continued fraction only contributing higher-order corrections. Therefore, matching the leading order of the $n=1$ ratio to the right-hand side gives $\sigma_1^2 \to 1$ as $d \to \infty$.

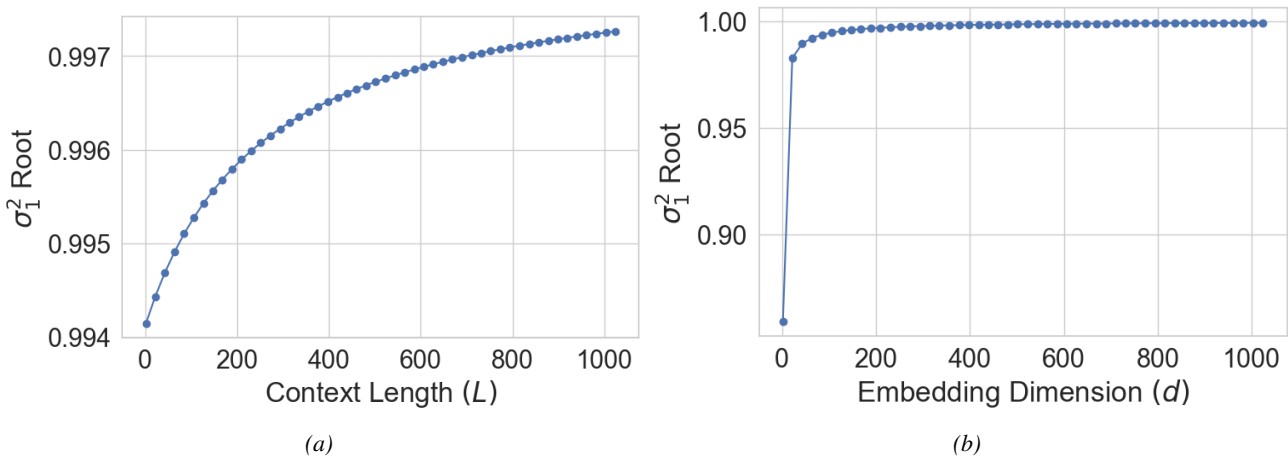

*Figure 8.* The stationary point found from Equation 19 for the shape parameter $\sigma_1$ of the lognormal proxy distribution as **(a)** a function of the context length $L$ with fixed embedding dimension $d = 128$ and as **(b)** a function of embedding dimension $d$ with fixed context length $L = 128$, by truncating the continued fraction at $N = 20$ terms.

## A.5. Statistical View of Temperature Scaling

Within this section, we investigate the conditions for the expectation of the $\log$ maximum component of the attention probability to be held constant. To do this, we use the spherically symmetric PDF for the $\mathrm{clr}(\mathbf{p})$ given in Equation 7, such that the radius and direction components are independent. Using this, we derive a necessary bound for the order of the inverse-temperature scaling function.

The structure of our argument is as follows: we take $\mathbb{E}[\log p_{\max}] \geq \log C$, where $C$ is constant for all context lengths $L$, as a lower bound, we then derive an upper bound $\mathbb{E}[\log p_{\max}] \leq f(L, T(L))$. Therefore, it is clear that we must have $f(L, T(L)) \geq \log C$, an inequality that we solve for $T(L)$, the inverse-temperature scaling function. Hence, this function order is a necessary condition for the lower bound condition to be possible.

To define our criterion for the temperature scaling, we use the notion that the expected $\log$ maximum probability $p_{\max}$ must remain at least constant as the context length $L$ increases. We can formally define this as the expression

$$\mathbb{E}[\log p_{\max}] \geq \log C, \quad \forall L > 1,$$

where we are using the inverse-temperature scaling function $T(L)$ on the attention logits $\mathbf{a}$ to give the probabilities, with context length $L$, as

$$p_j = \frac{\exp(T(L) \cdot a_j)}{\sum_{\ell=1}^{L} \exp(T(L) \cdot a_\ell)}.$$

For this bound we use the following definitions: the magnitude of the $\mathrm{clr}$ transformed probabilities, $\tilde{a} = \|\mathrm{clr}(\mathbf{p})\|$, the uniform directional component on the sphere within $H$, $\mathbf{n} = \frac{\mathrm{clr}(\mathbf{p})}{\|\mathrm{clr}(\mathbf{p})\|}$, and $m = \max_i \mathrm{clr}(\mathbf{p})_i$. We decompose this maximum $\mathrm{clr}$ coordinate $m$ such that this depends on $\mathbb{E}[\tilde{a}]$ and $\mathbb{E}[\max_i n_i]$.

Due to this spherical symmetry, we have the independence of the radius and direction components; hence, we can use the magnitude $\tilde{a}$ and the uniformity on the unit sphere $\mathbf{n}$ in $H$ to have

$$m = \max_i \mathrm{clr}(\mathbf{p})_i = \tilde{a} \max_i n_i,$$

which, in expectation terms, is given as the bound

$$\mathbb{E}[m] = \mathbb{E}[\tilde{a}] \cdot \mathbb{E}[\max_i n_i] \leq \mathbb{E}[\tilde{a}] \cdot \mathbb{E}[\max_i |n_i|].$$

**Magnitude** $\tilde{a}$ Using the moments generated from the MGF, with no inverse-temperature scaling function, i.e. $T(L) = 1$, the expectation of $\tilde{a}_{T(L)=1}$ is given by

$$\mathbb{E}[\tilde{a}_{T(L)=1}] = \frac{2\sqrt{c} \cdot \Gamma(\frac{d+1}{2}) \cdot \Gamma(\frac{L}{2})}{\Gamma(\frac{d}{2}) \cdot \Gamma(\frac{L-1}{2})},$$

which, when using the asymptotic limit of the gamma functions, gives an order of

$$\mathbb{E}[\tilde{a}_{T(L)=1}] = \mathcal{O}(\sqrt{L}).$$

However, due to the linear relationship between $\tilde{a}$ and the inverse-temperature scaling function, we have

$$\mathbb{E}[\tilde{a}] = T(L) \cdot \mathcal{O}(\sqrt{L}).$$

We can note here that this magnitude has a dependence on the embedding dimension $d$ and the normalisation $c$. With the standard normalisation $c = \frac{1}{d}$, this dependence on $d$ converges to a constant prefactor; therefore, we absorb this dependence and state the expectation of the $\tilde{a}$ component in the large $L$ limit.

**Uniform Sphere** $\max_i |n_i|$ Now we consider $\mathbb{E}[\max_i |n_i|]$. The $\text{clr}(\mathbf{p})$ vector is spherically symmetric on the hyperplane $H$, therefore, this uniform direction can be modelled as $\frac{P\mathbf{x}}{\|P\mathbf{x}\|_2}$, where $P$ is the centering projection matrix, and $\mathbf{x} \sim \mathcal{N}(\mathbf{0}, I_L)$. To bound the expectation of $\max_i |n_i| = \frac{\|P\mathbf{x}\|_\infty}{\|P\mathbf{x}\|_2}$, we use a probability tail bound for the numerator and denominator and integrate.

The centering projection $\mathbf{y} = P\mathbf{x}$ produces a correlated Gaussian distribution where each component has variance $1 - \frac{1}{L}$, hence, we use the standard probability bounds (Boucheron et al., 2013) for the following

$$\mathbb{P}(\|\mathbf{y}\|_\infty > t) \le 2L \exp(-\frac{t^2}{2(1 - \frac{1}{L})}).$$

Likewise, the denominator of the ratio can be expressed as $\|\mathbf{y}\|_2$, therefore, we have the distribution given as $\|\mathbf{y}\|_2^2 \sim \chi_{L-1}^2$ (Cochran, 1934), allowing the application of the bounds within Ghosh (2021) to give

$$\mathbb{P}\left(\|\mathbf{y}\|_2 < \sqrt{(1-\epsilon)(L-1)}\right) \le \exp(-\frac{\epsilon^2(L-1)}{4}),$$

where $\epsilon \in (0, 1)$ is fixed.

Hence, the probability tail of $\max_i |n_i|$ can be bounded as

$$\mathbb{P}\left(\frac{\|\mathbf{y}\|_\infty}{\|\mathbf{y}\|_2} > t\right) \le \mathbb{P}\left(\|\mathbf{y}\|_\infty > t\sqrt{(1-\epsilon)(L-1)}\right) + \mathbb{P}\left(\|\mathbf{y}\|_2 < \sqrt{(1-\epsilon)(L-1)}\right),$$

which we can integrate between 0 and 1, as $\frac{\|\mathbf{y}\|_\infty}{\|\mathbf{y}\|_2} \le 1$, to bound the expectation as

$$\mathbb{E}\left[\frac{\|\mathbf{y}\|_\infty}{\|\mathbf{y}\|_2}\right] \le \int_0^1 \min\left(1, 2L \exp(-\frac{(1-\epsilon)(L-1)t^2}{2(1 - \frac{1}{L})})\right) dt + \exp(-\frac{\epsilon^2(L-1)}{4}).$$

We can split this integral into two parts, where this is a valid probability and therefore below 1, by using $t_0 = \sqrt{\frac{2(1-\frac{1}{L})\log(2L)}{(1-\epsilon)(L-1)}} \le 1$, giving

$$\mathbb{E}\left[\frac{\|\mathbf{y}\|_\infty}{\|\mathbf{y}\|_2}\right] \le \sqrt{\frac{2(1 - \frac{1}{L})\log(2L)}{(1-\epsilon)(L-1)}} + \int_{t_0}^1 2L \exp(-\frac{(1-\epsilon)(L-1)t^2}{2(1 - \frac{1}{L})}) dt + \exp(-\frac{\epsilon^2(L-1)}{4}),$$

which, by integrating, gives an order for the expectation of the uniform direction in the $H$ hyperplane as

$$\mathbb{E}[\max_i |n_i|] = \mathcal{O}\left(\sqrt{\frac{\log L}{L}}\right), \tag{20}$$

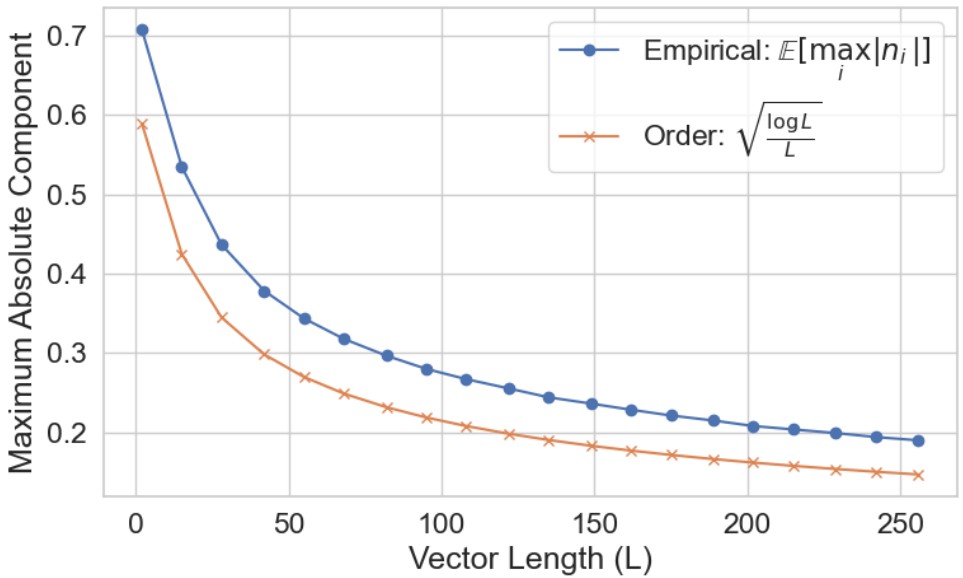

*Figure 9.* Empirical estimate for $\mathbb{E}[\max_i |n_i|]$, and the order that we bound in Equation 20.

which we can empirically verify in Figure 9.

Finally, combining this with the order of $\mathbb{E}[\tilde{a}]$, the expectation of $m$ has an order of

$$\mathbb{E}[m] = T(L) \cdot \mathcal{O}\left(\sqrt{\log L}\right).$$

**Probability Space**  Using this order, we now relate this back to the probability space by substituting our definitions back in to give the expressions

$$\mathbb{E}[m] = \mathbb{E}[\log \frac{p_{\max}}{g(\mathbf{p})}] = \mathbb{E}[\log p_{\max} - \log g(\mathbf{p})] = \mathbb{E}[\log p_{\max}] - \mathbb{E}[\log g(\mathbf{p})],$$

$$\Rightarrow \mathbb{E}[\log p_{\max}] = \mathbb{E}[m] + \mathbb{E}[\log g(\mathbf{p})].$$

The geometric mean $g(\mathbf{p})$ is bounded above by $\frac{1}{L}$, giving $\mathbb{E}[\log g(\mathbf{p})] \leq -\log L$. We also have the order bound of $\mathbb{E}[m] \leq T(L) \cdot C_1 \cdot \sqrt{\log L}$, where $C_1$ is a positive constant. Hence, we can simplify our problem as the loose bound

$$\mathbb{E}[\log p_{\max}] \leq T(L) \cdot C_1 \cdot \sqrt{\log L} - \log L. \tag{21}$$

Now, as this is an upper bound on the $\mathbb{E}[\log p_{\max}]$, and we have the lower bound condition as

$$\mathbb{E}[\log p_{\max}] \geq \log C, \tag{22}$$

for both Equation 21 and Equation 22 to be true, the upper bound must be above the lower bound, giving the bound

$$f(L, T(L)) = T(L) \cdot C_1 \cdot \sqrt{\log L} - \log L \geq \log C,$$

which, by solving this inequality for $T(L)$, gives

$$T(L) \geq \frac{\log C + \log L}{C_1 \cdot \sqrt{\log L}}.$$

Hence, our final order, necessary condition of $T(L)$ is given by

$$T(L) = \Omega\left(\sqrt{\log L}\right), \tag{23}$$

which is the function that inspires our second inverse-temperature scaling function seen within our token distinguishability real-world model experiments in Section 5. This function is a necessary condition for the constant expected $\log$ maximum probability to be possible.

To demonstrate this order scaling, we use a Gaussian query and key attention sampler and plot the maximum probability output with an increasing context length. As this is an order-bound, we manually tune a scalar $K$ value such that this inverse-temperature is given by $T(L) = K \cdot \sqrt{\log L}$, with the target to bound the maximum probability above an arbitrary $p_{\max} = 0.5$. We also plot the unscaled $p_{\max}$ for comparison, shown in Figure 10.

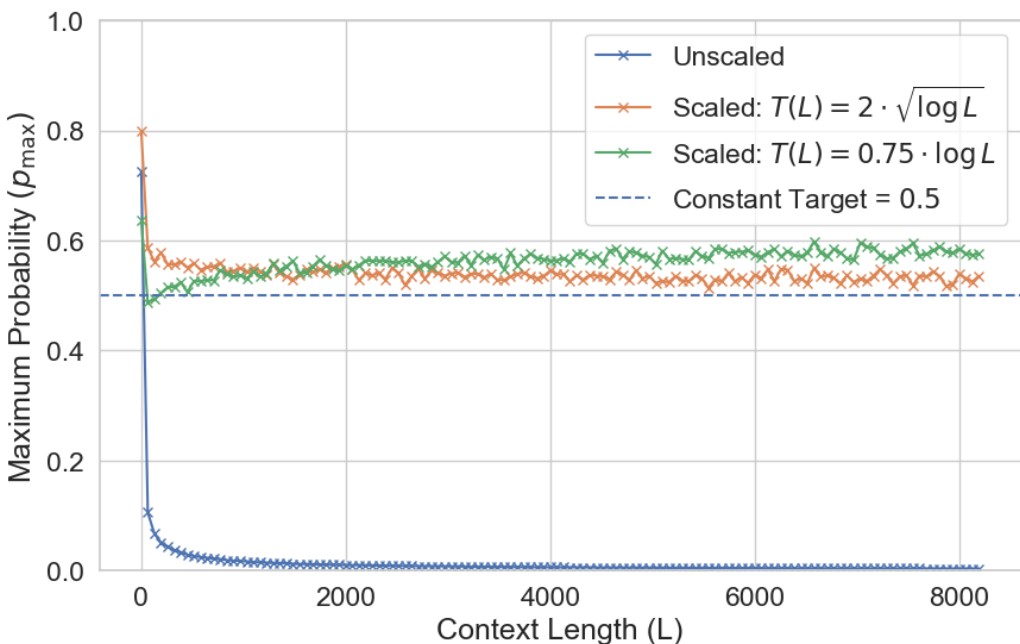

*Figure 10.* Empirical example, with $d = 32$, of the maximum probability from a Gaussian query key sampler, with and without the order scaling within Proposition 4.1 and the scaling found in Jianlin (2021); Chen et al. (2026), for context length up to $L = 8192$. This demonstrates the ability of a scaled $T(L) = K_1 \cdot \sqrt{\log L}$ function to keep the maximum probability approximately constant, where $K_1 = 2$, and $T(L) = K_2 \cdot \log L$, with $K_2 = 0.75$.

From Figure 10, we first observe the looseness of this lower bound, such that this has a slight negative trend in the max probability over the context length. Despite this, a function of order $T(L) = \mathcal{O}(\log L)$ appears to be large as the maximum probability is increasing with context length. Hence, this $\sqrt{\log L}$ scaling is a fair approximation for the scaling required with Gaussian queries and keys.

## B. Empirical Verifications

Within this section, we provide implementation details for the experiments within Section 5, and we expand and provide empirical verifications for the claims made within this paper. Firstly, in Section B.1, we plot and compare the full vector distribution of the lognormal proxy and the attention probabilities. Next, we provide the implementation details for the empirical use cases of Aitchison distance in Section B.2, with Section B.3 presenting the change in token distinguishability during training for different $\beta_0$ initialisation scales. Further, we extend our analysis of the length-extrapolated models in Section 5 to include another model pair and an analysis of the entropy weightings for KL divergence to uniform within Section B.4.

### B.1. Lognormal Approximation

Given our norm of the clr distribution matching of the attention probabilities distribution with the lognormal distribution in Section 3, we now compare the full vector distribution empirically.

We take 100 $L$-dimensional samples of the softmax of the Gaussian query and keys for the attention probabilities, and 100

$L$-dimensional samples of the lognormal distribution of size $L$, using $\mu = -\log L - \frac{1}{2}$, producing a mean of $\frac{1}{L}$, and the moment-matched shape parameter $\sigma_1 = 1$. For this, we use a fixed embedding dimension of $d = 512$, a fixed context length of $L = 512$, and plot the raw components histogram in Figure 11.

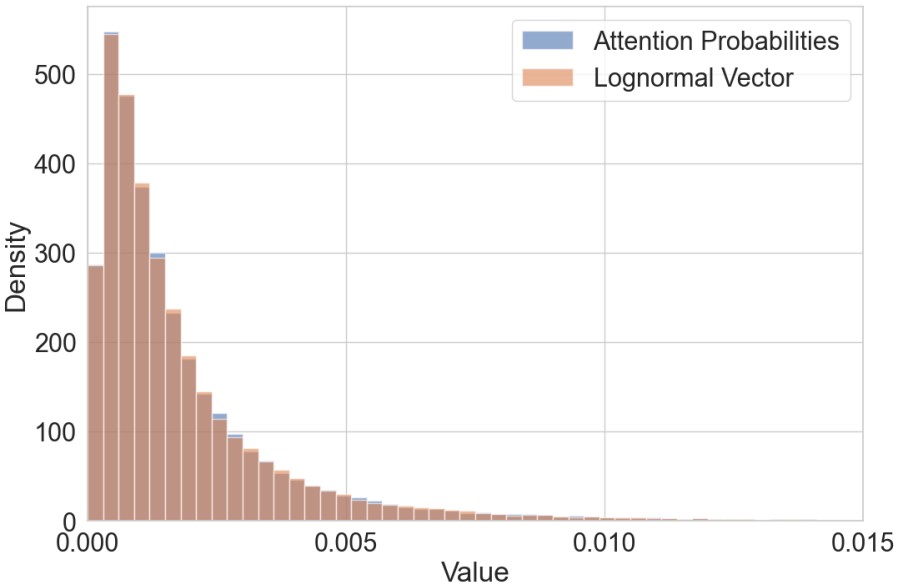

*Figure 11.* Comparison of the raw components of 100 samples of $L$-dimensional attention probabilities, generated by the softmax of the dot product of Gaussian query and key vectors, and lognormal distributed samples with $\sigma_1^2 = 1$, using an embedding dimension $d = 512$, and context length $L = 512$.

Figure 11 showcases the close match of these two distributions at a large embedding dimension and large context length. Hence, despite our analysis being limited to the 1D moment match of the shape parameter from the PDFs, following Nahshan et al. (2024), the lognormal distribution is a good fit for the distribution of Gaussian query and key attention probabilities.

## B.2. Implementation Details

In this section, we provide the implementation details and hyperparameters for the experiments presented within Section 5.

For each of these experiments, the code that we construct is a modification of the code from Zhang et al. (2024) and Lightning AI (2023). For the experiments that log distance measures of the attention probabilities during training, to allow the use of Flash-Attention2 (Dao, 2024), we implement a logging function to build the individual rows of the attention matrix of interest, i.e. Row 8 and 2048, and calculate the normalised Aitchison distance and normalised KL divergence for these rows. For the calculation of the Aitchison distance, we avoid taking the $\log$ of potentially small probabilities by directly calculating the norm of the centred attention logits. Hence, we use a standard transformer model with this additional logging and the registration of additional parameters within the temperature scaling experiment, outlined below, and optimising using AdamW (Loshchilov & Hutter, 2019).

### B.2.1. TOKEN DISTINGUISHABILITY WITH TEMPERATURE SCALING

For the investigation into the temperature scaling relation with token distinguishability, we train 4 Pythia-1B models, with the hyperparameters given in Table 1. These models are trained on the RedPajama-Data-1T-Sample 1B dataset with an additional $\beta = \text{ReLU}(\beta_{\text{raw}})$ parameter per layer. Hence, for the row of attention logits with length $L$, the logits are multiplied by $\beta \cdot T(L)$, where $\beta$ is initialised to $\beta_0 = 1.0$ for all layers. These 4 models have inverse-temperature scaling functions as $T(L) = \beta \cdot \{1, \sqrt{\log L}, \log L, (\log L)^2\}$.

### B.2.2. TOKEN DISTINGUISHABILITY WITH TRAINING REGIMES

For the experiments into the relation between training regime and token distinguishability, we use the same parameters as those in Table 1 for all 3 models, with the exception of the parameters in Table 2. All 3 Pythia-1B models were trained on

*Table 1.* Implementation parameters used for the training of the 4 Pythia-1B models, with an increasing order of inverse-temperature scaling from $T(L) = \beta \cdot \{1, \sqrt{\log L}, \log L, (\log L)^2\}$. The results of these models are presented in Figure 3 within Section 5.

| PARAMETER | PARAMETER SPECIFICATION |
|---|---|
| LEARNING RATE | $6 \times 10^{-4}$ |
| OPTIMISER | ADAMW |
| BATCH SIZE | 120 |
| MICRO BATCH SIZE | 6 |
| MAX ITERATIONS | $45,595$ |
| WEIGHT DECAY | 0.1 |
| GRADIENT CLIPPING | 1 |
| ADAMW: $\beta_1$, $\beta_2$ | $0.9, 0.95$ |
| LEARNING RATE DECAY MIN | $5.93 \times 10^{-4}$ |
| WARMUP ITERATIONS | 2000 |

the same RedPajama-Data-1T-Sample 1B dataset with an inverse-temperature scaling applied as $\beta \cdot \sqrt{\log L}$, where $\beta$ is learnable per layer, again initialised to $\beta_0 = 1.0$, and has been passed through the ReLU function.

*Table 2.* Changes in the parameters used in Table 1 for the training of the 3 Pythia-1B models: Healthy, Underfitting 1, Underfitting 2. Healthy uses standard hyperparameters for a sensible training regime, Underfitting 1 has a small learning rate and high weight decay, while Underfitting 2 has a high learning rate and no weight decay. The results of these models are presented in Figure 4 within Section 5.

| **HEALTHY** | |
|---|---|
| PARAMETER | PARAMETER SPECIFICATION |
| LEARNING RATE | $6 \times 10^{-4}$ |
| WEIGHT DECAY | 0.1 |
| LEARNING RATE DECAY MIN | $5.93 \times 10^{-4}$ |
| **UNDERFITTING 1** | |
| PARAMETER | PARAMETER SPECIFICATION |
| LEARNING RATE | $1 \times 10^{-6}$ |
| WEIGHT DECAY | 0.3 |
| LEARNING RATE DECAY MIN | $1 \times 10^{-6}$ |
| **UNDERFITTING 2** | |
| PARAMETER | PARAMETER SPECIFICATION |
| LEARNING RATE | $1 \times 10^{-2}$ |
| WEIGHT DECAY | 0.0 |
| LEARNING RATE DECAY MIN | $9.87 \times 10^{-3}$ |

### B.2.3. LENGTH EXTRAPOLATION DISTRIBUTION SHIFT

To compare the distribution shift when using a length-extrapolated model, we take 10 random samples of 512-token-length paragraphs from the RedPajama-Data-1T-Sample 1B dataset and use this as a prompt for the pre-trained models. During inference on all 10, we extract the causal attention matrix for all layers and heads. For the ridge plots, the $\log$ probabilities are pooled for all the heads in that layer, while for the distance measures of normalised Aitchison distance and normalised KL divergence, we plot the average over the heads and prompts for each row of these attention matrices.

### B.3. Temperature Scaling Initialisation Scales

To show that our findings about the difference in normalised Aitchison distance between context length rows holds for different initialisation scales, we conduct the same experiment for the various inverse-temperature scaling functions as in Section 5, and change the initialisation $\beta_0$ scale from $\beta_0 = 1.0$, seen in Figure 3, to $\beta_0 = 0.5$, in Figure 12. Likewise, following one of the initialisations within Nakanishi (2025), we use $\beta_0$ initialised to the reciprocal of the average inverse-temperature scaling function's value during training, i.e. $\beta_0 = \frac{2048}{\sum_{n=1}^{2048} T(n)}$, and plot the results in Figure 13.

From these experiments, we note that the initialisation of the scaling parameter plays a key role in the resultant magnitudes

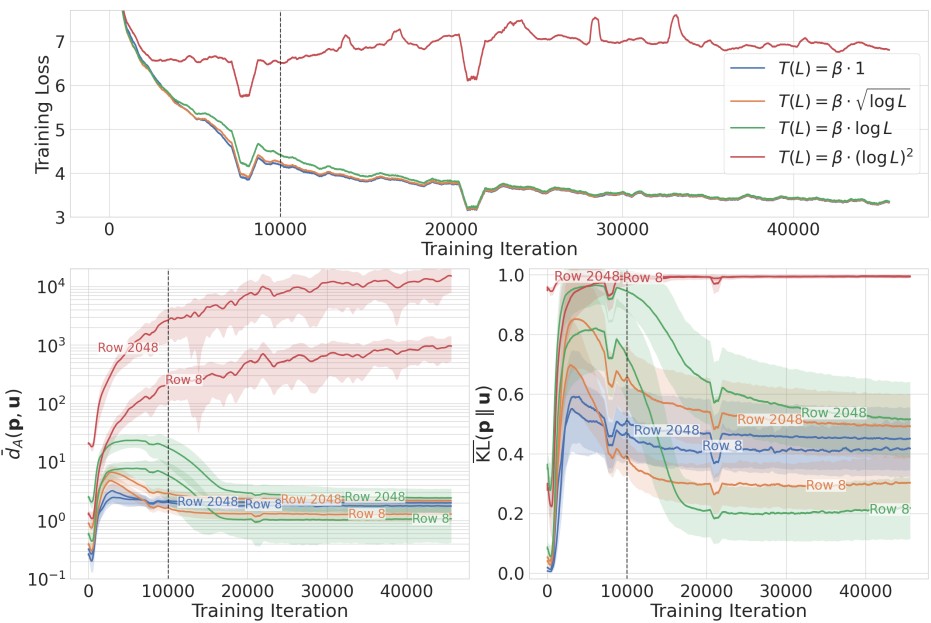

*Figure 12.* Training loss, normalised Aitchison distance, and normalised KL divergence to uniform for Row 8 and Row 2048 of the attention probabilities of a Pythia-1B model trained on the RedPajama-Data-1T-Sample 1B dataset, each with an inverse-temperature scaling function $T(L) = \beta \cdot \{1, \sqrt{\log L}, \log L, (\log L)^2\}$, where $\beta$ is initialised to $\beta_0 = 0.5$.

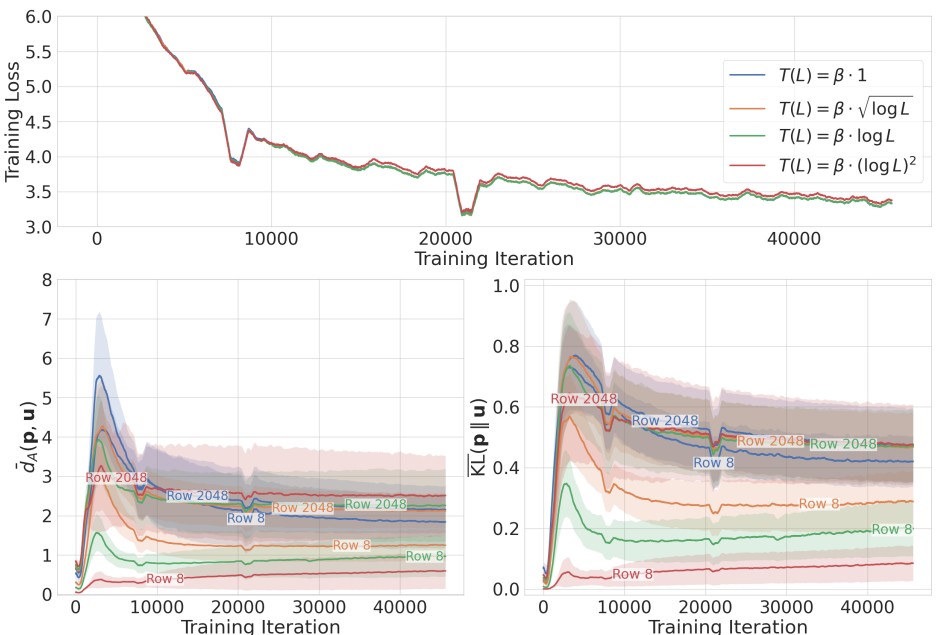

*Figure 13.* Training loss, normalised Aitchison distance, and normalised KL divergence to uniform for Row 8 and Row 2048 of the attention probabilities of a Pythia-1B model trained on the RedPajama-Data-1T-Sample 1B dataset, each with an inverse-temperature scaling function $T(L) = \beta \cdot \{1, \sqrt{\log L}, \log L, (\log L)^2\}$, where $\beta$ is initialised to $\beta_0 = \frac{2048}{\sum_{n=1}^{2048} T(n)}$.

of the distances to uniformity. In particular, when scaling each $\beta_0$ individually within Figure 13, a $T(L) = (\log L)^2$ scaling no longer automatically saturates the softmax and explodes the loss. However, our observations from Figure 3 remain; the difference in Aitchison distance between short context and long context rows scales with the order of the inverse-temperature scaling function, and a more selective pattern is required at long contexts, resulting in the distance to uniform of Row 2048 being larger than Row 8.

### B.4. Length-Extrapolated Pairs

To extend upon our analysis in Section 5, comparing the LongLLaMA-3B model and OpenLLaMA-3B model, we add the final memory layer, i.e. Layer 18, and we perform this comparison with a further length extrapolation model pair, LLaMA2-7B (Touvron et al., 2023b) and LLaMA2-7B-32k (Together AI, 2023).

#### B.4.1. LongLLaMA and OpenLLaMA Memory Layers

For completeness, we plot all of the memory layers, $\log$ probabilities and entropy weighting with the normalised Aitchison distance and KL divergence to uniform in Figure 14.

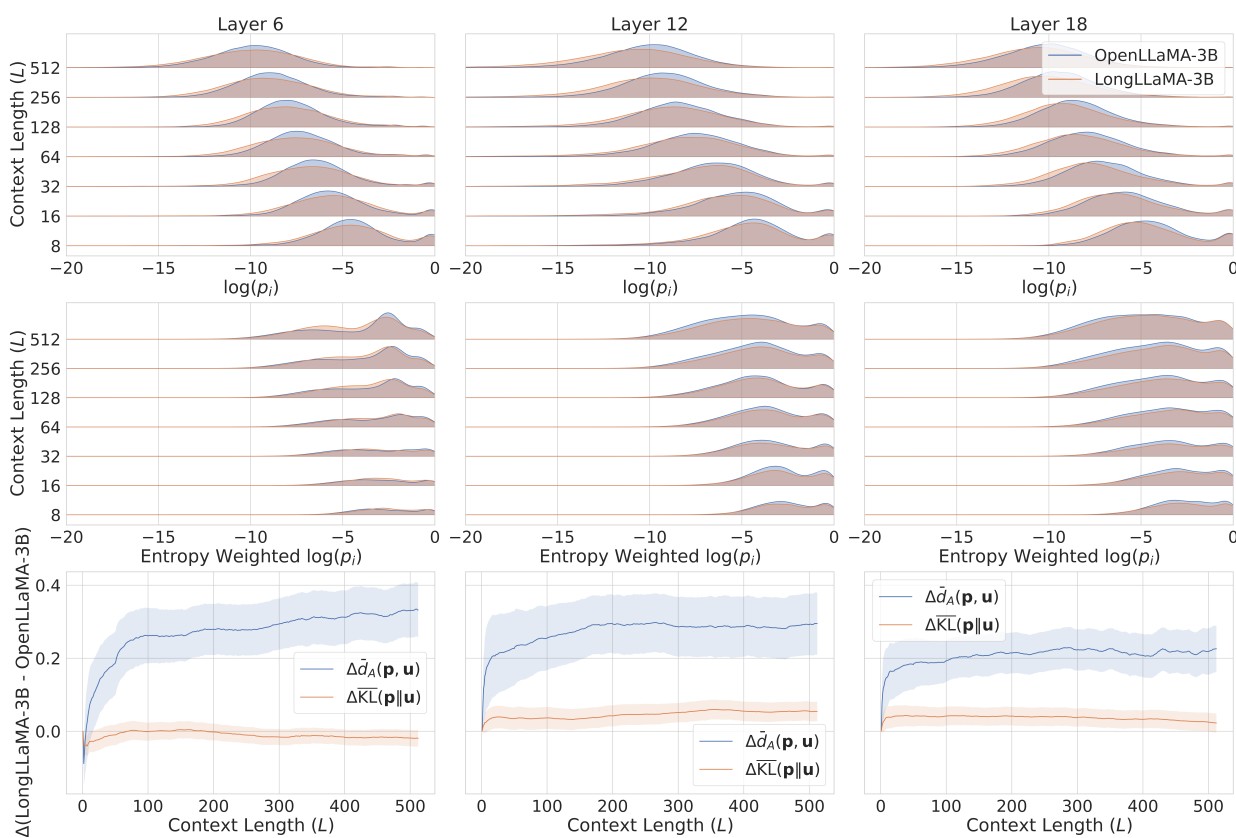

*Figure 14.* First Row: Ridge plot, showcasing the histogram of $\log p_i$ for an increasing context length of OpenLLaMA-3B and LongLLaMA-3B for all memory layers (Layer 6, 12, 18). Second Row: Entropy weighted $\log(p_i)$ ridge plot, weighting each sample with $-p_i \log p_i$, showing its contribution towards entropy measures. Third Row: The difference $\Delta$ between LongLLaMA-3B and OpenLLaMA-3B's normalised KL divergence and normalised Aitchison distance to uniform over the context length, averaged over all heads and over 10 random prompts of length 512 from the RedPajama-Data-1T-Sample 1B dataset, with the standard error for both measures.

Within the entropy weighted ridge plots of Figure 14, we can observe that, despite the small shifts in the larger probability components, the weighting within entropy measures increases these components' importance, overlooking the importance of many smaller components. Take Layer 12 as an example, the $\log$ probabilities show that many smaller components have been shrunk, making this attention more focused; however, within the entropy weighting, these smaller tokens make a small to no contribution towards the normalised KL measure. Therefore, when we want to consider the full vector of probabilities, Aitchison distance is more suitable.

#### B.4.2. LLaMA-2 Model Comparison

We also compare the models Llama-2-7B and the length-extrapolated model Llama-2-7B-32k in Figure 15. While this does not explicitly use memory layers, we still plot the 6th, 12th and 18th layer, such that a similar trend as seen in the LongLLaMA-3B and OpenLLaMA-3B occurs.

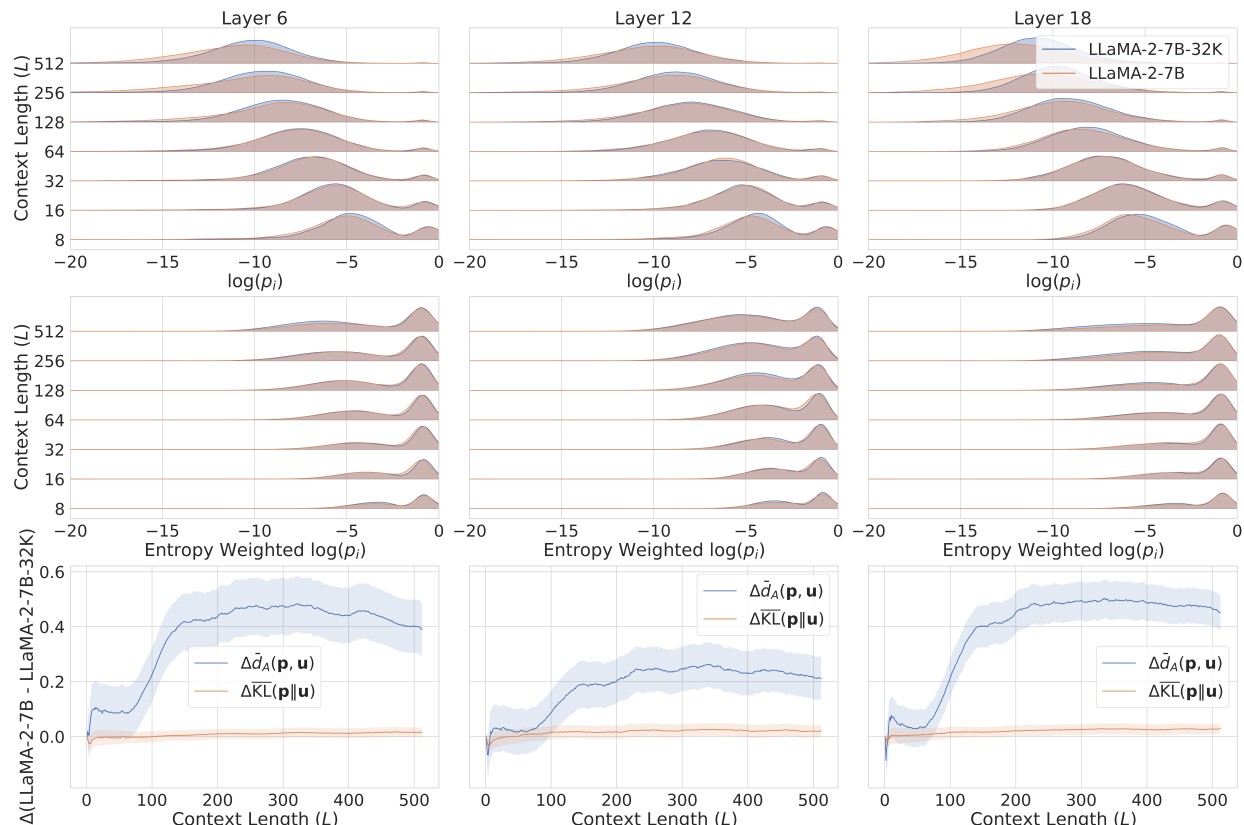

*Figure 15.* First Row: Ridge plot, showcasing the histogram of $\log p_i$ for an increasing context length of Llama-2-7B and Llama-2-7B-32k for the layers 6, 12, and 18. Second Row: Entropy weighted $\log(p_i)$ ridge plot, weighting each sample with $-p_i \log p_i$, showing its contribution towards entropy measures. Third Row: The difference $\Delta$ between Llama-2-7B-32k and Llama-2-7B's normalised KL divergence and normalised Aitchison distance to uniform over the context length, averaged over all heads and over 10 random prompts of length 512 from the RedPajama-Data-1T-Sample 1B dataset, with the standard error for both measures.

Within all of the layers that we plot, there is a significant reduction in the weight of the smallest probability values. This corresponds with a large Aitchison distance to uniform increase between the length-extrapolated models and base models, with the largest shifts occurring within the 6th and 18th layers. Despite this large shift in the smaller components, the difference between the KL divergence to uniform between the layers is smaller, indicating the lower sensitivity to the smaller components of the attention probabilities.

Hence, unlike the example of the normalised Aitchison distance and KL divergence to uniform disagreeing in the LongLLaMA-3B and OpenLLaMA-3B models, the distance measures on Llama-2-7B-32k and Llama-2-7B agree on the direction of uniformity for these layers. Despite this, Aitchison distance clearly separates the magnitude of this shift, i.e. a larger shift in the probabilities occurs in Layers 6 and 18 when compared to Layer 12.

