# OpenReview forum: "Modelling Attention with Aitchison Geometry: Token Distinguishability and Temperature Scaling"
_ICML.cc/2026/Conference — ICML 2026 regular_

### Official Review · Reviewer_EgSV · 2026-03-11

**Soundness:** 4
**Presentation:** 4
**Significance:** 4
**Originality:** 4
**Overall Recommendation:** 5
**Confidence:** 5

**Summary:**

This manuscript provides a formal theoretical characterization of token distinguishability in the Transformer attention mechanism using Aitchison geometry. The authors address the concern of "vanishing attention" (softmax dispersion) in long contexts, where individual probabilities shrink as sequence length $L$ increases. By introducing the Aitchison distance—a metric from Compositional Data Analysis (CoDA)—the study demonstrates that the relative differences among attention probabilities (log ratios) do not vanish but converge to a finite, non-zero limit. The paper's notable contribution pertains to the derivation of a theoretical lower bound of $\Omega(\sqrt{\log L})$ for temperature scaling to maintain a sharp attention distribution. Overall, this paper discusses an important concept that establishes Aitchison distance as a principled alternative to entropy for monitoring model state during training and inference.

**Compliance With Llm Reviewing Policy:**

Affirmed.

**Key Questions For Authors:**

1. Proposition 4.1 derives a $\Omega(\sqrt{\log L})$ lower bound. How do the authors reconcile this with concurrent works that suggest a $\log L$ scaling for entropy preservation?
2. The analysis assumes independent Gaussian queries and keys. How would the limits of distinguishability change if queries and keys are correlated or living on a low-rank manifold?
3. Can the authors propose a practical way to integrate Aitchison distance into a standard loss function for pre-training to explicitly preserve tail distinguishability?

**Limitations:**

The analysis is primarily conducted at initialization. While the authors present training curves in Section 5, a formal theoretical characterization of Aitchison distance dynamics during the learning process remains a future challenge.

**Strengths And Weaknesses:**

Soundness:
The technical analysis is highly sophisticated and mathematically rigorous. The authors derive the exact probability density function (PDF) for the Aitchison distance to uniform under the assumption of Gaussian queries and keys (Proposition 3.1) and successfully calculate the Moment Generating Function (MGF). The verification of the $\Omega(\sqrt{\log L})$ scaling via probability tail bounds for the uniform sphere is particularly impressive. The empirical verification of the lognormal approximation for attention distributions adds an additional layer of soundness to the theoretical model.
Presentation:
The paper is excellently written and structured. Figure 1 effectively illustrates why Aitchison distance is superior to KL divergence for capturing changes in smaller probability components. The ridge plots in Figure 5 and Figure 12 demonstrate the practical diagnostic utility of the measure when comparing base models with length-extrapolated versions like LongLLaMA.
Significance:
The significance of this work is foundational. As context windows in models like GPT-4 and Gemini extend to millions of tokens, the theoretical limits of attention selectivity become critical. This work provides the first rigorous mathematical proof that distinguishability persists in the limit, offering a theoretical justification for existing temperature scaling heuristics in models like YaRN. The introduction of Aitchison distance as a diagnostic tool for "representational collapse" is a major step forward for model observability.
Originality:
The application of Aitchison geometry—a tool typically used in geology and chemistry—to the attention mechanism is a highly original and creative interdisciplinary synthesis. The derivation of the linear relationship between temperature and Aitchison distance offers new insights into the mechanism of logit scaling.

---

> ### Author Rebuttal · Authors · 2026-03-31
>
> Thank you for the feedback and review of our work.
>
> # Questions
> ## **Q1**
> Thank you for this question, highlighting two concurrent works in particular [1, 2], a $\sqrt{\log L}$ and $\log{L}$ temperature scale is derived in each. As discussed within [2], the discrepancy between these two findings is the modelling assumption that changes the gap between the top pairwise inner products. In the geometric model given in [2], this gap remains $\mathcal{O}(1)$, whereas within the Random Energy Model proposed in [1], this gap shrinks with $(1/\sqrt{\log L})$ scale.
>
> Our work more closely follows the modelling assumptions of [1], which takes the attention logits to be normally distributed. Instead, we model the queries and keys as separate Gaussian vectors, which form the distribution of the Aitchison distance given in Eq. 1. By separating these queries and keys, we can track the dependence of this distance on the head dimension as well as the context length.
>
> ## **Q2**
> This modelling assumption of independent Gaussian query and keys enables a tractable analysis to isolate the impact of the softmax's sum-to-one constraint.
>
> Regarding your question, we can take the case of uniformly correlated queries and keys with unit variance and consider the norm of the centred attention logits, or Aitchison distance. With $\mathbf{q} \sim \mathcal{N}(0, I_d)$ and $\mathbf{k}\_i = \rho \mathbf{q} + \sqrt{1-\rho^2} \boldsymbol{\epsilon}\_{i}$ where $\boldsymbol{\epsilon}\_i \sim \mathcal{N}(0, I_d)$, i.e. the queries and keys are correlated with correlation $\rho$. Then for $a_i = \mathbf{q}\mathbf{k}\_i^{\top}/\sqrt{d}$, we have $a_i - \bar a = \frac{1}{\sqrt{d}} \sqrt{1-\rho^2} \mathbf{q}  (\boldsymbol{\epsilon}\_i - \boldsymbol{\bar \epsilon})^{\top}$, where $\boldsymbol{\bar \epsilon} = \frac{1}{L}\sum_{i=1}^L \boldsymbol{\epsilon}\_i$ and $\bar a = \frac{1}{L}\sum_{i= 1}^{L} a_i$. Hence, the norm of the centred attention logits is $||\mathbf{\bar a}|| = \sqrt{1-\rho^2}||\frac{1}{\sqrt{d}} \mathbf{q} E^{\top}||$, where $E$ is the matrix obtained by stacking the centred $\boldsymbol{\epsilon}\_i - \boldsymbol{\bar \epsilon}$ vectors. Therefore, the Aitchison distance has been rescaled with $\sqrt{1-\rho^2}$.
>
> For the low rankness, taking the simple case where the $d$ dimensional queries and keys are determined by vectors of $r$ dimensions using the matrix $A \in \mathbb{R}^{r \times d}$, i.e. $\mathbf{q} = \mathbf{z_q} A, \mathbf{k}\_i = \mathbf{z}\_{\mathbf{k}\_i}A $, with $\mathbf{z_q}, \mathbf{z_{k_i}} \sim \mathcal{N}(0, \phi_{qk}^2 I_r)$, and $A$ having orthonormal rows, a semi-orthogonal matrix. Then the attention logits are given as $a_i = \frac{\mathbf{z_q} \mathbf{z}\_{\mathbf{k}\_i}^{\top}}{\sqrt{d}}$, as $A A^{\top} = I_r$. Hence, the distribution given in Eq. 1 has all of the $d$ values changed to $r$, but with the $c = \frac{\phi_{qk}^4}{d}$ normalisation factor remaining the same. Therefore, the resultant expected normalised Aitchison distance in the limit is given as $\lim_{L \rightarrow \infty} \mathbb{E}[\bar d_A(\mathbf{p}, \mathbf{u})] = \frac{\sqrt{2c} \cdot \Gamma(\frac{r+1}{2})}{\Gamma{(\frac{r}{2})}}$, which is rescaled version of the independent limit, but does not collapse in the limit of the context length.
>
> ## **Q3**
> Thank you for the suggestion. While we have not considered this direction within this work, motivated by this question, we have developed a regularisation method that adds a penalty to the cross-entropy loss that scales with the Aitchison distance.
>
> As Aitchison distance can be large, we use a penalty of the form $-\lambda \cdot \overline{\log(1 + \bar d_A(\mathbf{p}, \mathbf{u})^2)}$ with $\lambda > 0$, where this penalty is averaged over all layers and heads. In using this, we observe a similar trend to that seen in the comparisons between the length extrapolated models in Fig. 5 and Fig. 13, where the spread of the attention probabilities increases when performing inference between the non-regularised and regularised models. Hence, this regularisation penalty has increased the distance to uniformity of the attention probabilities succesfully and the model is more "selective". However, due to space limitations and the short rebuttal period, we have not been able to benchmark the models' capabilities adequately. In particular, the regularisation strength parameter would need to be tuned for specific tasks. We will aim to include a discussion of the practical implementations with Aitchison distance in the revised version.
>
> ## **L1**
> As per Reviewer bCW9's comments, we agree that research investigating the evolution of token distinguishability would be a valuable addition in future works.
>
> # References
> [1] Giorlandino, A. and Goldt, S. Two failure modes of deep transformers and how to avoid them: a unified theory of signal propagation at initialisation. ICLR 2026
>
> [2] Chen, S., Lin, Z., Polyanskiy, Y., and Rigollet, P. Critical attention scaling in long-context transformers. ICLR 2026.

---

### Official Review · Reviewer_bCW9 · 2026-03-11

**Soundness:** 3
**Presentation:** 3
**Significance:** 3
**Originality:** 2
**Overall Recommendation:** 5
**Confidence:** 3

**Summary:**

This paper thoroughly examines the significance of replacing entropy with Aitchson distance for training monitoring in attention-based models. It elaborates on the sensitivity of Aitchson distance to low-probability components and provides detailed reasoning behind this capability. Aitchson distance offers developers effective recommendations for attention resource allocation strategies, enabling targeted adjustments to training parameters and strategies across different layers. The paper features clear logic, a well-structured writing process, reasonable experiments, and rigorous conclusions.

**Compliance With Llm Reviewing Policy:**

Affirmed.

**Final Justification:**

the author has addressed my concern

**Key Questions For Authors:**

- As mentioned in the weakness section, I would like the author to briefly analyze and elaborate on potential scenarios where “exceptions” might occur in the temperature scaling rules, along with their associated impacts and corresponding countermeasures. The content need not be extensive—a short paragraph will suffice—primarily to enhance the completeness of the main text.

- Theoretically, can this approach be applied to analyze all types of contexts? Can the same method be used for image tasks, 3D tasks, and text analysis tasks?

**Limitations:**

See weakness

**Strengths And Weaknesses:**

### Strengths:

- The authors introduce a principled and innovative theoretical framework grounded in Aitchison geometry, effectively pioneering the integration of Compositional Data Analysis (CoDA) into the formal characterization of attention mechanisms. By conceptualizing attention probability distributions as discrete coordinates residing within a simplex space, this study leverages log-ratios—as opposed to absolute numerical magnitudes—to quantify the relative disparities between tokens. Such a methodology establishes a geometric perspective that is inherently commensurate with the intrinsic mathematical properties of the attention mechanism, specifically its sum-to-one constraint, thereby providing a more rigorous and theoretically sound foundation for analysis.
- The framework demonstrates a superior sensitivity to the long-tail information of attention distributions, providing a more exhaustive evaluation of distributional shifts than traditional metrics such as information entropy or Kullback-Leibler (KL) divergence. Whereas KL divergence is characteristically biased toward variations in dominant probability components, the Aitchison distance maintains a high degree of sensitivity to fluctuations within infinitesimal tail components. In the context of long-sequence modeling, where focusing is realized through the systematic suppression of negligible probabilities across a multitude of irrelevant tokens, the Aitchison distance effectively quantifies these subtle structural reorganizations to which conventional divergence-based metrics are largely insensitive.
- The work establishes a rigorous theoretical foundation for temperature scaling by exploiting the intrinsic linear relationship between Aitchison distance and the scaling factor. Specifically, the authors derive a necessary lower bound for the temperature scaling function, requisite for maintaining the sharpness of attention distributions within long-context regimes. This contribution provides a principled scientific framework for the configuration of hyperparameters in long-context architectural design, thereby effectively mitigating the necessity for heuristic-based empirical tuning.

### Weakness

- The primary limitation is that this study examines the properties of attention models in static states, with insufficient research on models undergoing continuous change during training. This is particularly critical if Aitchson distance is to serve as a monitoring tool guiding training. However, the authors clearly acknowledge this issue and commit to addressing it in future research. Moreover, the analysis presented in the main text holds practical significance as a theoretical framework. Therefore, I consider this limitation acceptable.

- The main text mentions that the derived lower bound for temperature shrinkage is neither necessary nor sufficient for the training objective, implying that exceptions may occur during immediate application. While I understand that discussing these exceptions might deviate from the paper's core focus, I believe the author should briefly address—perhaps within a single paragraph—the probability of such exceptions occurring and the practical engineering steps we should take to avoid them when applying the lower bound. This addition would, in my view, enhance the completeness of the main text.

---

> ### Author Rebuttal · Authors · 2026-03-31
>
> We very much appreciate the Reviewer's feedback on our work.
>
> # Weaknesses
> ## **W1**
> Thank you for your detailed view on this limitation. We agree that more research would be valuable into the evolution of this token uniformity during training.
>
> To make a step in this direction, we have studied the empirical evolution of LLMs on language modelling tasks. For example, Fig. 3 and 4 demonstrate that token distinguishability does not vanish at longer contexts when viewing this through the Aitchison distance lens.
>
> # Questions
> ## **W2, Q1**
> As discussed on Line 315 of our paper, the lower bound that we derive is a **necessary lower bound**. Therefore, while this temperature scale does not guarantee that the maximum attention probability remains above $0$ in the long context regime, this states that the temperature scale must be at least as big as $\sqrt{\log L}$ for it to be possible to avoid this collapse.
>
> Moreover, as in Fig. 10 in the Appendix, this temperature scale on a toy Gaussian query and key model up to $8192$ context length results in a very slow negative trend in the maximum probability component, with the $\log L$ scale being large enough to scale the maximum probability component up with context length. Hence, the temperature scale to keep this maximum probability constant is between $\sqrt{\log L}$ and $\log L$. Despite this, with this slight decrease in the maximum probability with $\sqrt{\log L}$, a very large context length would be required to collapse these probabilities to $0$, motivating the use of $\sqrt{\log L}$ as the temperature scale in our practical experiments. To extend on this, we have increased the context length in Fig. 10 to $128,000$, which shows that even at this very long context, the $\sqrt{\log L}$ scaling maintains probability mass on the largest component above the $0.5$ threshold that we use. We will include this extended figure in our revised version.
>
> ## **Q2**
> In this work, we study the general attention mechanism. In particular, we study the impact of an increasing context length on the attention probabilities. Therefore, any task that requires the model to perform attention over a large "context", whether this be image patches or word tokens, has this simplex sum-to-one restriction. Hence, temperature scaling of this order would still apply to prevent the collapse of these probabilities towards the uniform attention, degenerate regime.

---

> > ### Author Rebuttal · Reviewer_bCW9 · 2026-04-03
> >
> > Good, i'll keep my "accept" rating

---

### Official Review · Reviewer_vKYp · 2026-03-12

**Soundness:** 4
**Presentation:** 4
**Significance:** 3
**Originality:** 3
**Overall Recommendation:** 4
**Confidence:** 4

**Summary:**

This paper studies token distinguishability in Transformer attention under long-context regimes. The authors analyze the behavior of attention probabilities using Aitchison geometry, viewing the softmax outputs as compositional data on the probability simplex. Under Gaussian assumptions for queries and keys, the paper derives asymptotic results showing that token distinguishability, measured via the Aitchison distance, converges to a finite non-zero limit as the context length grows. Based on this analysis, the paper further derives a theoretical lower bound of \Omega(\sqrt{\log L}) for temperature scaling required to maintain sharp attention distributions, which theoretically explain the temperature scaling of \log L in real applications.

**Compliance With Llm Reviewing Policy:**

Affirmed.

**Final Justification:**

The author solved my concerns.

**Key Questions For Authors:**

Could the authors clarify whether the key conclusions depend critically on the use of Aitchison geometry? In particular, would similar scaling laws for temperature emerge if token distinguishability were measured using alternative metrics such as entropy, KL divergence, or logit variance?



**Overall, I appreciate this line of theoretical work.**

**Limitations:**

Yes

**Strengths And Weaknesses:**

**Strengths:**

(1) The theory is interesting and it explains the temperature selection in practical applications.

(2)  The paper is well-written, and problem setting is clear. Readers can easily follow the paper and catch the main message.


**Weaknesses:**

(1) The main conclusions regarding temperature scaling and attention dispersion are closely related to previous theoretical analyses of softmax attention under large context regimes. I am not sure how much theoretical contributions and technical difficulties and novelty, compared with previous works achieving similar theoretical results.

(2) The analysis relies on strong assumptions such as Gaussian queries and keys, which may not accurately reflect the behavior of trained Transformer models with normalization layers, positional encodings, and residual connections. Can this assumption be further relaxed? Or does the real model approximately satisfy this assumption?

(3) The empirical validation is relatively limited. The experiments mainly consist of synthetic or simplified settings, and the paper does not convincingly demonstrate whether the theoretical insights hold in real large-scale language models.

---

> ### Author Rebuttal · Authors · 2026-03-31
>
> Thank you very much for the detailed review and feedback.
>
> # Weaknesses
> ## **W1**
> You are right that this area of research has received significant attention recently. While temperature scales of order $\sqrt{\log L}$ and $\log L$ have been theoretically derived in prior works such as [1, 2], our primary novelty is to introduce a principled and more comprehensive modelling approach, i.e. for the first time to apply Aitchison distance to compare the attention distribution and the
> uniform distribution to characterise token distinguishability.
>
> Then, based on this novel modelling, **1)** we derive the temperature scaling as a lower bound. Additionally, **2)** we found that token distinguishability is non-vanishing with context length; however, in practice, probabilities can underflow to $0$, motivating our investigation in linking this token distinguishability with temperature scaling to prevent this.
>
> Compared to existing works, all these derived results are novel and provide new insights and findings for this area.
>
> More specifically, our work more closely follows the modelling assumptions of [1], where they assume correlated Gaussian attention logits. In contrast, we split the query and keys into separate Gaussian vectors and statistically analyse the token distinguishability of the probabilities by relating the softmax application to Aitchison geometry. Additionally, by separating the query and keys, we can track the impact of the head dimension on the distribution of the token distinguishability.
>
> ## **W2**
> We agree that Gaussian query and key vectors do not exactly match the true behaviour of trained Transformers. The scope of this paper is to study the static behaviour of the attention mechanism; we do this by assuming a Gaussian initialisation and investigating the impact that the softmax has on the resultant attention probabilities in this "neutral state".
>
> Further, the distribution assumption can be relaxed with the following: in using Aitchison distance to measure token distinguishability, as in Eq. 4, the variance of the attention logits is equal to the normalised Aitchison distance squared. Hence, if any distribution $\mathcal{P}$ of the attention logits has a non-vanishing variance in context length, the normalised Aitchison distance squared is also non-vanishing with context length. Likewise, in empirically investigating the change in the token distinguishability during a real model's training, we show that Aitchison distance does not vanish with context length on language modelling tasks.
>
> ## **W3**
> Unfortunately, we can't train a real large scale language model during the short rebuttal period. However, we aim to complete a pre-training of an LLM with >1B parameters in the revised version.
>
> More generally, our work focuses on a theoretical characterisation of a general attention mechanism; therefore, we anticipate that these insights hold for a variety of sized models and tasks. Further, we have aimed to supplement the LLM pretraining by using Aitchison distance as a tool to measure distinguishability during inference on much larger models up to 7B parameters.
>
> # Questions
> ## **Q1**
> Thank you for raising this important point. Introducing Aitchison geometry is due to its capability to measure the log ratios of the probabilities to investigate the token distinguishability, defined as the ability for the model to pick between tokens (not collapsing towards the same values). This is exactly the main contribution of our paper. As motivated in Sec. 2 of our paper, while metrics like entropy/KL divergence are informative about the sharpness of the distribution, these do not equally weight the importance of all probabilities, making smaller probabilities less impactful on the closeness to uniformity. As shown in Eq. 4, logit variance is equal to the normalised Aitchison distance squared; therefore, using the logit variance would give equivalent results to those of the Aitchison distance.
>
> For the temperature scaling, the primary focus is on the scale required for it to be possible for the maximum probability not to collapse towards $0$ in the long-context regime, and therefore, for the model to retain some form of token distinguishability. This part does not critically depend on the use of Aitchison distance. Our derivation uses this measure due to its concrete link with the statistical distribution of the norm of the centred attention logit. Therefore, while our method is closely tied to Aitchison distance, as seen by other works, similar temperature scales can be derived through statistical physics methods [1] and by identifying the contractive properties of the softmax operator [2].
>
> # References
> [1] Giorlandino, A. and Goldt, S. Two failure modes of deep transformers and how to avoid them: a unified theory of signal propagation at initialisation. ICLR 2026
>
> [2] Chen, S., Lin, Z., Polyanskiy, Y., and Rigollet, P. Critical attention scaling in long-context transformers. ICLR 2026.

---

> > ### Author Rebuttal · Reviewer_vKYp · 2026-04-03
> >
> > Thank you for the response. The authors have addressed my comments.

---

### Official Review · Reviewer_ErkB · 2026-03-13

**Soundness:** 3
**Presentation:** 1
**Significance:** 3
**Originality:** 3
**Overall Recommendation:** 4
**Confidence:** 2

**Summary:**

This paper addresses the problem of "vanishing attention" (softmax dispersion) in Transformers as context lengths scale. By treating attention probabilities as compositional data, the authors provide both theoretical bounds and practical solutions for long-context models by introducing the normalized Aitchison distance, which measures token distinguishability by analyzing the log-ratios between attention probabilities rather than their raw, diminishing magnitudes. It proves to be far more sensitive to "tail" probability shifts than standard entropy metrics (like KL divergence), as demonstrated on LLaMA-2 and LongLLaMA. The paper mathematically proves (under Gaussian query/key assumptions) that the relative distinguishability of tokens does not vanish, but rather converges to a non-zero constant as context length $L \to \infty$. It also derives a rigorous statistical lower bound, proving that a temperature scaling function of at least $\Omega(\sqrt{\log L})$ is required to maintain a constant log maximum probability. This theoretical baseline was successfully validated during the training of Pythia-410m models.

**Compliance With Llm Reviewing Policy:**

Affirmed.

**Final Justification:**

I thank the authors for their comprehensive response. The additional clarifications have sufficiently addressed my primary concerns. In light of these improvements, I have raised my score to Weak Accept.

**Key Questions For Authors:**

1. How do you reconcile your theoretical non-vanishing limit for relative distinguishability with real-world hardware (FP16/INT8), where diminishing absolute probabilities will inevitably underflow to zero?

2. Isn't the idealized Aitchison geometry claim entirely dependent on the $\Omega ( \log L)$ temperature scaling to prevent this underflow? Shouldn't this critical physical constraint be explicitly stated alongside your main theoretical limit?

3. Your temperature scaling was validated on Pythia-410m. Do you expect the $\Omega ( \log L)$ bound to hold strictly for much larger state-of-the-art models (>7B) or alternative architectures like Grouped Query Attention?

4. The heavy reliance on the appendix disrupts the paper's flow. Which specific proofs or explanations do you plan to move into the main text to improve readability?

**Limitations:**

yes

**Strengths And Weaknesses:**

- `The Scale-Invariance Flaw and Hardware Quantization`: While the Aitchison distance captures relative token distinguishability, its scale-invariance masks a critical physical limitation. Because the metric only evaluates the log-ratios between tokens, it ignores the reality that as context length $L \to \infty$, the absolute magnitude of every individual probability shrinks proportionally to $1/L$.
Consequently, the paper's foundational theoretical claim—that relative distance remains constant at infinite context—is physically meaningless in isolation. In real-world hardware utilizing lower-precision formats (FP16, BF16, or INT8), these microscopic absolute values inevitably underflow to exactly zero, utterly destroying the "relative distinguishability" (as one cannot compute the log-ratio of zero). Therefore, the entire theoretical framework is strictly dependent on the temperature scaling proposed in Section 4. Without the $\Omega(\sqrt{\log L})$ scaling to rescue the maximum absolute probabilities ($\mathbb{E}[\log p_{max}]$) from machine underflow, the idealized Aitchison geometry would immediately collapse in practice.

- `Presentation and Readability`: The paper suffers from structural issues that significantly hamper readability. The authors frequently defer crucial mathematical steps, proofs, and contextual explanations to the appendix. This forces the reader to constantly flip back and forth, disrupting the narrative flow and making the core theoretical arguments unnecessarily difficult to follow in the main text.

---

> ### Author Rebuttal · Authors · 2026-03-31
>
> Firstly, we would like to thank the reviewer for the careful review and feedback on our work.
>
> # Questions
> As the first two questions are related, we will answer both of them at once.
> ## **Q1**, **Q2**
> You are exactly right regarding the underflow of attention probabilities to $0$; we claim that tokens maintain their distinguishability up to machine precision. Therefore, while our analysis states that this distinguishability is maintained at infinite context length, when using an infinite precision regime, in practice, underflowing to $0$ is a real concern for very large contexts. This is a major motivation for us to explore the temperature scaling to avoid this collapse of the maximum attention probability. In deriving the necessary condition, we show that, under our assumptions, maintaining probabilities above $0$ in the long context regime is not possible without a temperature scale of order $\Omega(\sqrt{\log L})$.
>
> Therefore, while our theoretical investigation studies the idealistic conditions of infinite machine precision and infinite context length, we provide theoretically-grounded practical guidance for the real-world conditions where we would want to avoid token uniformity by suggesting temperature scaling.
>
> > In real-world hardware utilizing lower-precision formats (FP16,
> > BF16, or INT8), these microscopic
> > absolute values inevitably underflow to exactly zero, utterly
> > destroying the "relative distinguishability" (as one cannot
> > compute the log-ratio of zero).
>
> This raises a good point relating to the calculation of the token uniformity. As highlighted within Sec. 2, Aitchison distance is equivalent to the norm of the centred attention logits; therefore, when tracking the Aitchison distance, we use the attention logits rather than the probabilities to avoid this underflow issue. Hence, we can still calculate the measure of token distinguishability without taking the $\log$ of $0$ or using some arbitrary $\epsilon >0$ threshold. This also means that this tracker cannot detect if numerical underflow is occurring after the softmax; however, the purpose of the tracker is to measure the underlying token concentration. In the case where we are interested in the numerical underflow issues, a separate tracker to measure the number of exact $0$s would be more suitable, paired with this Aitchison distance.
>
> > Shouldn't this critical physical constraint be explicitly stated alongside your main theoretical limit?
>
> Thank you for the suggestion to make explicitly clear this connection between the real-world machine precision, our theoretical non-vanishing results and temperature scaling. To address this, we will add a paragraph before our main contributions discussing the link between Sections 3, 4 and real-world machine precision.
>
> ## **Q3**
> Training bigger sizes of LLMs is unfortunately not feasible for us during the short rebuttal period. Despite this, we will aim to train a larger model (>1B parameters) for the revised version. Additionally, our theory focuses on characterising the token uniformity and temperature scaling when using a general attention mechanism; therefore, we anticipate that this applies to a wide variety of model sizes.
>
> Just as with a normal attention mechanism, Grouped Query Attention (GQA) produces an attention logit vector for each query; therefore, these logits can be scaled with a temperature scale just as with a standard attention mechanism. Hence, with our modelling assumptions, our theory applies to the general attention mechanism; therefore, the constraint of summing-to-one probabilities that the softmax imposes could apply to a wide range of architectures or implementations that use the softmax on an increasingly sized "context", including GQA.
>
> ## **Q4**
> Thank you for the suggestion. To improve the readability of this work, we propose including a proof sketch and explanation of the key findings, Proposition 4.1 in particular. Likewise, as discussed above, we propose an additional paragraph discussing the relation of the non-vanishing theory with real-world machine precision and temperature scaling.
>
> Finally, we propose making the implementation and purpose of the Aitchison distance tracker more explicit within Sec. 5 in response to the discussion around taking the $\log$ of $0$.

---

> > ### Author Rebuttal · Reviewer_ErkB · 2026-04-01
> >
> > I thank the authors for their comprehensive response. The additional clarifications have sufficiently addressed my primary concerns. In light of these improvements, I am happy to raise my score to Weak Accept.

---

### Decision · Program_Chairs · 2026-04-30

**Decision:**

Accept (regular)

**Comment:**

This paper studies long-context attention through the lens of compositional data analysis. The paper's notable contribution pertains to introducing Aitchison geometry as a principled way to measure token distinguishability in softmax attention, showing under Gaussian query-key assumptions that normalized distinguishability does not vanish with context length, deriving a theoretical lower bound of $\Omega(\sqrt{\log L})$ for temperature scaling, and demonstrating that Aitchison distance is useful in practice for monitoring training and inference. Overall, this paper discusses an important concept in long-context transformers: how to reason about distinguishability when raw attention probabilities shrink with sequence length.

I recommend acceptance. The paper makes a clear conceptual contribution by reframing attention probabilities as compositional objects and introducing a metric that is better aligned with relative token distinguishability than standard entropy-based summaries. The theory is technically meaningful, the $\Omega(\sqrt{\log L})$ lower bound is well motivated, and the empirical sections support practical relevance by showing how Aitchison distance tracks training dynamics and captures changes in attention structure that KL is less sensitive to.

The concerns raised by reviewers were mostly about presentation, assumptions, and breadth of validation rather than flaws in the core contribution, and the rebuttal addressed these points satisfactorily. In particular, the authors clarified the relationship to finite-precision underflow, explained how Aitchison distance is computed from centered logits in practice, and better positioned the role of the theory and its assumptions. Reviewers explicitly indicated that their concerns were resolved and maintained or raised their scores accordingly.

Overall, this is a solid and original ICML paper: the key idea is elegant, the analysis is useful, and the proposed perspective is likely to be valuable for future work on long-context attention and scaling.